# A Review of Bayesian Uncertainty Quantification in Deep Probabilistic Image Segmentation

**Amaan Valiuddin**                                    *m.m.a.valiuddin@tue.nl*
**Ruud van Sloun**                                     *r.j.g.v.sloun@tue.nl*
**Christiaan Viviers**                                 *c.g.a.viviers@tue.nl*
**Peter de With**                                      *p.h.n.de.with@tue.nl*
**Fons van der Sommen**                                *fvdsommen@tue.nl*
*Department of Electrical Engineering*
*Eindhoven University of Technology*
*Eindhoven, The Netherlands*

**Reviewed on OpenReview:** *https://openreview.net/forum?id=Yzf4anYwao*

## Abstract

Advances in architectural design, data availability, and compute have driven remarkable progress in semantic segmentation. Yet, these models often rely on relaxed Bayesian assumptions, omitting critical uncertainty information needed for robust decision-making. Despite growing interest in probabilistic segmentation to address point-estimate limitations, the research landscape remains fragmented. In response, this review synthesizes foundational concepts in uncertainty modeling, analyzing how feature- and parameter-distribution modeling impact four key segmentation tasks: Observer Variability, Active Learning, Model Introspection, and Model Generalization. Our work establishes a common framework by standardizing theory, notation, and terminology, thereby bridging the gap between method developers, task specialists, and applied researchers. We then discuss critical challenges, including the nuanced distinction between uncertainty types, strong assumptions in spatial aggregation, the lack of standardized benchmarks, and pitfalls in current quantification methods. We identify promising avenues for future research, such as uncertainty-aware active learning, data-driven benchmarks, transformer-based models, and novel techniques to move from simple segmentation problems to uncertainty in holistic scene understanding. Based on our analysis, we offer practical guidelines for researchers on method selection, evaluation, reproducibility, and meaningful uncertainty estimation. Ultimately, our goal is to facilitate the development of more reliable, efficient, and interpretable segmentation models that can be confidently deployed in real-world applications.

## 1 Introduction

Image segmentation entails pixel-wise classification of data, effectively delineating objects and regions of interest (Szeliski, 2010). The advent of convolutional neural networks (CNNs) has led to major breakthroughs in this domain (Ronneberger et al., 2015; Shelhamer et al., 2014; Badrinarayanan et al., 2015), obtaining impressive scores with large-scale segmentation datasets (Lin et al., 2014; Cordts et al., 2016b; Richter et al., 2016). However, these models typically rely on strong assumptions and significant relaxations of the Bayesian learning paradigm, neglecting the uncertainty associated within their predictions. This lack of uncertainty modeling reduces both the reliability and interpretability of these predictions. In high-stake applications, such as autonomous driving or medical diagnosis, this can have severe consequences. For instance, misclassifying objects in autonomous driving or overlooking uncertainty in lesion classification can both lead to critical decision-making errors.

Fortunately, the merits of uncertainty quantification have been well-recognized in the field of CNN-based segmentation, especially as interpretability and reliability have become central in data-driven applications.

Hence, extensive efforts have been made to align neural network optimization with Bayesian machine learning, such as learning parameter distributions rather than point estimates or to explicitly model the likelihood distribution of model features (Blundell et al., 2015; Guo et al., 2017; Kingma & Welling, 2013; Kendall & Gal, 2017). At the same time, uncertainty in image segmentation often treated as an auxiliary tool to enhance downstream performance, rather than as a primary modeling objective. Consequently, many studies lack a rigorous theoretical foundation, prioritizing empirical gains over principled formulations and clear definitions of uncertainty. This narrow focus frequently results in methods tailored to specific datasets or modalities, which limits their generalizability. Due to this application-driven nature, most existing surveys adopt a medical perspective (Jungo & Reyes, 2019; Kwon et al., 2020), often focused on specific modalities (McCrindle et al., 2021; Jungo et al., 2020; Ng et al., 2022; Roshanzamir et al., 2023). Additionally, the study by Kahl et al. (2024) introduces a valuable framework for benchmarking uncertainty disentanglement in semantic segmentation, but focuses primarily on empirical evaluation within specific tasks. Moreover, theoretical contributions often originate from adjacent fields like Bayesian deep learning or information theory, and can be insufficiently contextualized within the domain of segmentation. This disconnect has resulted in a fragmented body of work, with inconsistent terminology, evaluation metrics, and assumptions. The resulting abundance of literature can be overwhelming, even for experienced researchers.

Given the fragmented nature of the existing literature and the renewed interest in uncertainty (Papamarkou et al., 2024; Kirchhof et al., 2025), we note that a comprehensive overview in the field remains limited. Our central contribution is to unify core uncertainty modeling concepts across methods, tasks, and applications, thereby offering substantial theoretical insights and demonstrating their relevance across diverse domains. We unify and contextualize the underlying theory, and standardize terminology and notation. We systematically compare methodologies, link them to specific tasks, and connect those tasks to practical applications. Hence, this review is not only a conventional "zero-to-hero" survey; rather, it additionally serves as a cross-domain synthesis that bridges method developers, task specialists, and application-focused researchers. Our aim is to provide a cohesive framework for navigating and integrating the complex uncertainty modeling landscape. This broader perspective helps clarify conceptual foundations, highlight cross-cutting insights and provide a comprehensive grasp of the key challenges and open research directions in the field. In the absence of broad quantitative consensus, we adopt a robust, evidence-based methodology for making recommendations. By leveraging a deep qualitative analysis of the literature, we derive reliable method comparisons and guidelines that offer essential and practical guidance. However, this reliance on qualitative synthesis rather than quantitative evidence is an important limitation and our recommendations should be interpreted accordingly.

We begin by establishing the background on image segmentation in Section 2. Then, Section 3 introduces the notation and general theoretical principles of probabilistic image segmentation. Subsequently, we organize the review based on the source of introduced stochasticity: the feature level (Section 4) and the parameter level (Section 5). This structure aligns with the underlying theoretical decomposition of uncertainty and the utility of the modeled uncertainty for the downstream tasks and application domains discussed in Section 6. The review concludes by moving to the discussion (Section 7), where we highlight important pitfalls, challenges and opportunities in the field. This is followed by recommendations and guidelines for researchers (Section 8), before concluding in Section 9. The core structure and organizational flow of this paper is visualized in Figure 1, which serves as a top-level roadmap for the reader.

## 2 Background

A common approach in image understanding involves labeling pixels according to semantic categories, which is the core of *semantic segmentation*. This technique is particularly well-suited for amorphous or uncountable subjects. In contrast, *instance segmentation* not only assigns class labels, but also distinguishes and delineates individual object occurrences. This makes it more appropriate for scenarios involving countable entities. A third variant, *panoptic segmentation*, unifies both semantic and instance-level labeling, offering a comprehensive view of scene composition. As summarized by Minaee et al. (2020), semantic segmentation has been performed using methods such as thresholding (Otsu, 1979), histogram-based bundling, region-growing (Dhanachandra et al., 2015), k-means clustering (Nock & Nielsen, 2004), watershedding (Najman & Schmitt, 1994), to more advanced algorithms such as active contours (Kass et al., 2004), graph cuts (Boykov et al., 2001), conditional and Markov random fields (Plath et al., 2009), and sparsity-based methods (Starck

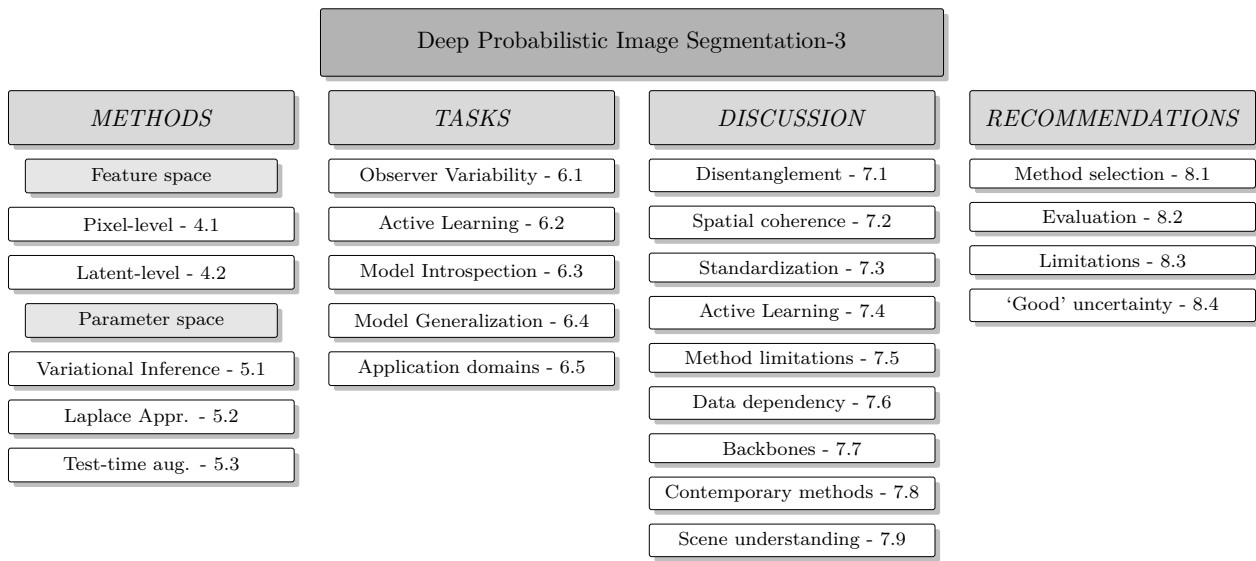

Figure 1: Overview of the sections. The leafs of the presented hierarchical tree are related to subsections of the article.

et al., 2005; Minaee & Wang, 2017). While the literature on segmentation is vast and rapidly evolving, a selection of backbone architectures has been particularly influential in shaping current probabilistic segmentation models. The focus here is not on exhaustiveness, but on those models most relevant to the development of uncertainty-aware approaches.

In particular, following the successful application of CNNs (LeCun et al., 1998), image segmentation experienced rapid progress driven by increasingly powerful and specialized deep architectures. Notably, the Fully Convolutional Network (FCN) (Shelhamer et al., 2014) adapted the AlexNet (Krizhevsky et al., 2012), VGG16 (Simonyan & Zisserman, 2014) and GoogLeNet (Szegedy et al., 2014) architectures to enable end-to-end semantic segmentation. Furthermore, other CNN architectures such as DeepLabv3 (Chen et al., 2017), and the MobileNetv3 (Howard et al., 2019) have also been commonly used. As the research progressed, increasing success has been observed with encoder-decoder models (Noh et al., 2015; Badrinarayanan et al., 2015; Yuan et al., 2019; Ronneberger et al., 2015). Initially developed for the medical applications, Ronneberger et al. (2015) introduced the U-Net, which successfully relies on residual connections between the encoding-decoding path, to preserve high-frequency details in the encoded feature maps. To this day, the U-Net is still often utilized as the default backbone model for many semantic segmentation and even general image generation architectures (Ho et al., 2020; Song et al., 2020), particularly in the medical domain. In fact, reports of recent research indicate that the relatively simple U-Net (Isensee et al., 2020) still outperform more contemporary and complex models (Eisenmann et al., 2023; Isensee et al., 2024).

## 3 Probabilistic Image Segmentation

Assuming random-variable pairs $(\mathbf{Y}, \mathbf{X}) \sim P_{\mathbf{Y}, \mathbf{X}}$ that take values in $\mathcal{Y} \in \mathbb{Z}^{K \times H \times W}$ and $\mathcal{X} \in \mathbb{R}^{C \times H \times W}$, respectively, then instance $\mathbf{y}$ can be considered as the ground-truth of a $K$-class segmentation task and instance $\mathbf{x}$ as the query image. The variables $H$, $W$ and $C$ correspond to the image height, width and channel depth, respectively. Conforming to the principle of maximum entropy, the optimal parameters given the data (i.e. posterior) subject to the chosen intermediate distributions can be inferred through Bayes Theorem as

$$p(\boldsymbol{\theta}|\mathbf{y}, \mathbf{x}) = \frac{p(\mathbf{y}|\mathbf{x}, \boldsymbol{\theta})p(\boldsymbol{\theta})}{p(\mathbf{y}|\mathbf{x})}, \tag{1}$$

assuming independence of $\mathbf{x}$ from $\boldsymbol{\theta}$, and where $p(\boldsymbol{\theta})$ represents the prior belief on the parameter distribution and $p(\mathbf{y}|\mathbf{x})$ the conditional data likelihood (also commonly referred to as the *evidence*). After obtaining a

posterior with dataset $\mathcal{D} = \{\mathbf{x}_i, \mathbf{y}_i\}_{i=1}^N$ containing $N$ images, the predictive distribution for a new datapoint $\mathbf{x}^*$ can be denoted as

$$p(\mathbf{Y}|\mathbf{x}^*, \mathcal{D}) = \int \underbrace{p(\mathbf{Y}|\mathbf{x}^*, \boldsymbol{\theta})}_{\text{Data}} \overbrace{p(\boldsymbol{\theta}|\mathcal{D})}^{\text{Model}} \, \mathrm{d}\boldsymbol{\theta}. \tag{2}$$

As evident, both the variability in the empirical data and the inferred parameters of the model influence the predictive distribution. A straightforward approach to quantify the total uncertainty is achieved by obtaining the predictive entropy, $H[\mathbf{Y}|\mathbf{x}^*, \mathcal{D}]$, i.e. the entropy of the predictive distribution $p(\mathbf{Y}|\mathbf{x}^*, \mathcal{D})$, as

$$H[\mathbf{Y}|\mathbf{x}^*, \mathcal{D}] = \overbrace{I[\mathbf{Y}, \boldsymbol{\theta}|\mathbf{x}^*, \mathcal{D}]}^{\text{epistemic}} + \underbrace{\mathbb{E}_{q(\boldsymbol{\theta}|\mathcal{D})}[H[\mathbf{Y}|\mathbf{x}^*, \boldsymbol{\theta}]]}_{\text{aleatoric}}, \tag{3}$$

where $I$ represents the mutual information. The first term represents the expected information gain about the model parameters $\boldsymbol{\theta}$ given the true label $\mathbf{Y}$. A high value indicates the model is very uncertain about its prediction, meaning observing the true label would provide a large information gain. The second term represents the inherent randomness in the data, found by averaging the predictive uncertainty across the posterior of plausible models from Equation (1). This reflects the noise that cannot be reduced, regardless of how much more training data is observed. This distinction allows us to determine if a model's uncertainty stems from its own ignorance or from ambiguity inherent in the data. Therefore, uncertainty is often split into two types: *epistemic* uncertainty, which reflects the model's limited knowledge of its parameters, and *aleatoric* uncertainty, which captures the irreducible statistical randomness in the data itself. Nonetheless, determining the nature of uncertainty is rarely straightforward. For example, Hüllermeier & Waegeman (2021) stated that "*by allowing the learner to change the setting, the distinction between these two types of uncertainty will be somewhat blurred*". This sentiment is also shared by Der Kiureghian & Ditlevsen (2009), noting that "*in one model an addressed uncertainty may be aleatory, in another model it may be epistemic*". Sharing similar views, we highlight the necessity of careful analyses and possible subjective interpretation regarding the topic as we treat the realm of quantifying spatially correlated uncertainty. Especially with increasingly complex methodologies, treating the uncertainties as separate concepts is mostly theoretical (often even more philosophical) and highly non-trivial in practice.

### 3.1 Conventional segmentation

Regardless of the elegantly formulated Bayesian posterior, most practical approaches make use of so-called "deterministic" segmentation networks. Such models are trained by Maximum Likelihood Estimation (MLE), specified as

$$\boldsymbol{\theta}_{\text{MLE}} = \arg \max_{\boldsymbol{\theta}} \log p(\mathbf{y}|\mathbf{x}, \boldsymbol{\theta}), \tag{4}$$

which simplifies the training procedure by taking a point estimate of the posterior. This approximation improves as the training data increases and the model parameter variances approach zero. As such, MLE

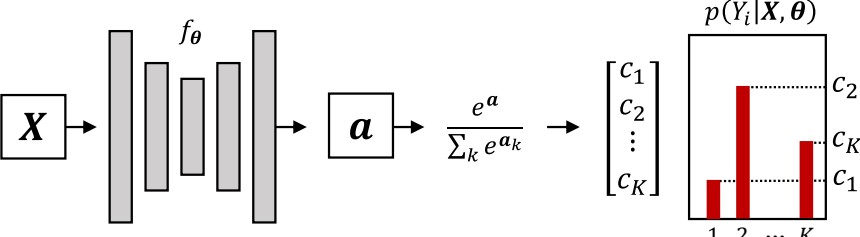

Figure 2: Uncertainty modeling by treating pixel-level confidence values as parameters of a Probability Mass Function.

does not include any prior knowledge on the structure of the parameter distribution. This, however, can be achieved through Maximum A Posteriori (MAP) estimation with

$$\boldsymbol{\theta}_{\text{MAP}} = \arg\max_{\boldsymbol{\theta}} \log p(\mathbf{y}|\mathbf{x}, \boldsymbol{\theta}) + \log p(\boldsymbol{\theta}). \tag{5}$$

For example, assuming Gaussian or Laplacian priors leads to regularizing the $L_2$ norm (also known as *ridge regression* or *weight decay*) or $L_1$ norm of $\boldsymbol{\theta}$, respectively (Figueiredo, 2001; Kaban, 2007). To model $p(\mathbf{y}|\mathbf{x}, \boldsymbol{\theta})$, we make use of a function $f_{\boldsymbol{\theta}} : \mathbb{R}^{C \times D} \to \mathbb{R}^{K \times D}$ that infers the parameters of a Probability Mass Function (PMF). For instance, consider spatial image dimensions $D$ and a CNN with $\mathbf{a} = f_\theta(\mathbf{x})$. Then, we can write

$$p(\mathbf{Y} = k \,|\, \mathbf{x}^*, \boldsymbol{\theta}) = \frac{e^{\mathbf{a}_k}}{\sum_k e^{\mathbf{a}_k}}, \tag{6}$$

with channel-wise indexing over the denominator, which is commonly known as the SoftMax activation (Figure 2). Hence, this approach is probabilistic modeling in the technical sense, although it is not referred to as such in common nomenclature. As such, the approximated distribution can represent and localize uncertain regions. However, the implicit pixel-independence assumption

$$p(\mathbf{Y}|\mathbf{X}) = \prod_i^{K \times D} p(Y_i|\mathbf{X}), \tag{7}$$

omits information on structural variation in the segmentation masks. In conventional classification, factorizing the categorical distribution is typically regarded as a logical simplification. In probabilistic segmentation however, this assumption has caused the emergence of a distinct research direction (See Figure 3).

> **Key takeaways**: Conventional models yield independent uncertainty estimates for each pixel, which omits spatial coherence. This failure to model spatial correlation precludes realistic inference and inhibits discerning common underlying causes for uncertainty in neighboring regions.

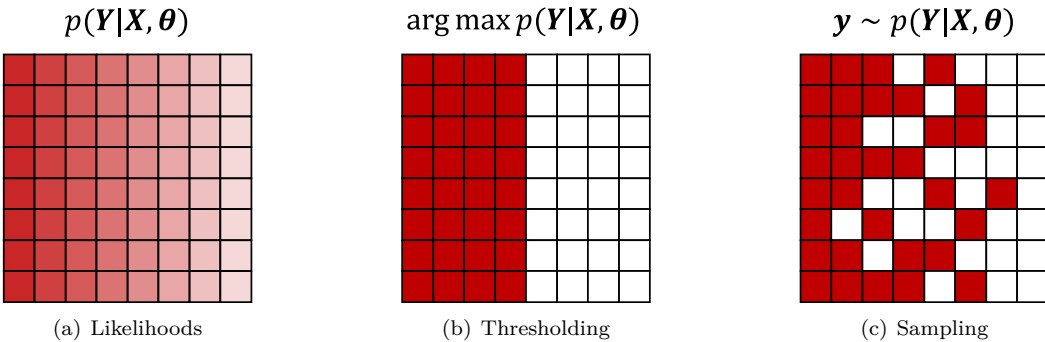

$$p(Y|X,\boldsymbol{\theta}) \qquad \arg\max p(Y|X,\boldsymbol{\theta}) \qquad y \sim p(Y|X,\boldsymbol{\theta})$$

(a) Likelihoods        (b) Thresholding        (c) Sampling

Figure 3: Illustration of the likelihood function in segmentation models. Color intensities reflect the normalized confidence values that can be interpreted as probabilities. (a) The continuous output results in a horizontal gradient. (b) Maximum likelihood thresholding can be applied. (c) However, the coherence of the segmentation suffers when sampling because information on neighboring pixels is disregarded.

## 3.2 Review organization and structure

The body of this review is structured around three interconnected concepts that serve as the fundamental organizing framework for the field of uncertainty in image segmentation. This structure is intended to facilitate cross-domain synthesis and enable researchers from diverse backgrounds to navigate the literature effectively. Consistent with the decomposition in Equation 2, we structure the methods based on feature-level (Section 4) and parameter-level (Section 5) uncertainty modeling. We then detail the utility of these

uncertainties for four key downstream tasks in Section 6, linking them to methods via Tables 1–4, where the connection to practical application domains is summarized in Table 5. Following this structured overview, we provide a detailed discussion (Section 7) to synthesize key insights, culminating in practical recommendations for the field (Section 8).

---

**Core concepts**:

- **Methods**: How uncertainty is mathematically modeled and quantified.
- **Tasks**: What specific goals are accomplished using these uncertainty estimates.
- **Applications**: Where these methods and tasks are applied in real-world domains.

---

## 4 Feature modeling

Following Equation (2), we focus in this section on approaches that place uncertainty at the level of features. In most practical scenarios, methods assume a parameterized likelihood function $p(\mathbf{Y}|\mathbf{X}, \boldsymbol{\theta})$, where parameters are point estimates. Accordingly, the central design decision is where to introduce auxiliary stochastic variables. Hence, we categorize models based on the location of these latent variables: those introduced near the output are discussed in Section 4.1, while lower-dimensional or latents embedded deeper within the architecture are covered in Section 4.2. Although their theoretical formulations are similar, the practical consequences can differ substantially.

### 4.1 Pixel-level sampling

Uncertainty in segmentation masks can be modeled directly at the pixel level. These approaches can be further categorized into those that assume independence between pixels (Section 4.1.1), and those that explicitly model spatial correlations (Section 4.1.2). In the former case, accurate uncertainty estimates rely on well-calibrated models, which is an assumption that often fails in practice without additional tuning on a separate calibration set. Furthermore, such methods do not address the challenge of spatial coherence. In contrast, the latter approach introduces a stochastic variable to capture dependencies between neighboring pixels.

### 4.1.1 Independence

While it is well established that assuming independence across pixels is problematic, it was a standard approach in much of the early literature. As discussed demonstrated in Equation (6), neural network predictions are often normalized with SoftMax activation in order to interpret the confidence values as parameters of a probability mass distribution. However, these values rarely reflect the true probabilities in modern neural networks, hypothesized due to the use of negative log-likelihood as the training objective, along with regularization techniques such as batch normalization, weight decay, and others (Guo et al., 2017). Thus, interpreting the confidences as true probabilities is often only justified after proper validation, which is referred to as *model calibration*.

**Calibration**   Different methods can be used to quantify whether the empirical accuracy of a model approximately equals the provided class confidence $c_k$ for class $k$, i.e. $P(Y = k \,|\, c_k) = c_k$. A fairly straightforward metric, the Expected Calibration Error (ECE), determines the normalized distance between accuracy (acc) and confidence (conf) bins as

$$E_{\text{ECE}} = \sum_{b=1}^{B} \frac{n_b}{N} |\text{acc}(b) - \text{conf}(b)|, \tag{8}$$

with $n_b$ the number of samples in bin $b$ and $N$ being the total sample size across all bins. The ECE is prone to skew representations if some bins are significantly more populated with samples due to over-/under-confidence. Furthermore, the Maximum Calibration Error (MCE) is more appropriate for high-risk applications, where only the worst bin is considered. Additionally, when background pixels have a

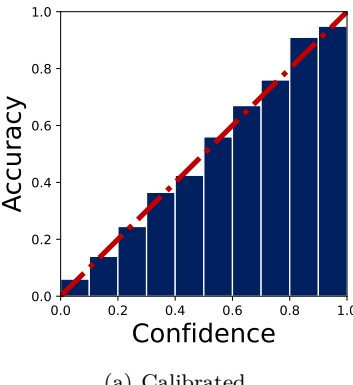
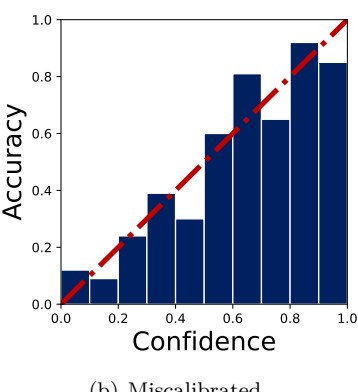

(a) Calibrated  (b) Miscalibrated

Figure 4: Visualization of model calibration with a reliability diagram, which demonstrates whether the predicted model confidences align with the observed empirical accuracies.

predominant influence, each bin can be weighted equally using the Average Calibration Error (ACE) to mitigate this imbalance (Jungo et al., 2020; Neumann et al., 2018). Calibration is typically visualized with a reliability diagram, where miscalibration and under-/over-confidence can be assessed by inspecting the deviation from the graph diagonal (Figure 4).

**Methods**  Most of the calibration techniques are post-hoc, i.e. applied after training, they necessitate a separate validation set. For example. over-fitting has often been considered to be the cause of over-confidence (Szegedy et al., 2016; Pereyra et al., 2017) and erroneous pixels can therefore be penalized through regularizing low-entropy outputs (Larrazabal et al., 2021a). Another prominent method is Temperature Scaling (Guo et al., 2017), which has been successfully applied in a pixel-wise manner to segmentation problems (Ding et al., 2021). The standard softmax probability in Equation (6) is modified as

$$p(\mathbf{Y} = k \,|\, \mathbf{x}^*, \boldsymbol{\theta}) = \frac{e^{\mathbf{a}_k/T}}{\sum_k e^{\mathbf{a}_k/T}}, \tag{9}$$

with some scalar $T$ optimized on a validation set. For overconfident models, $T > 1$ softens the estimated distribution, pushing output probabilities away from the extremes of 0 and 1 toward more moderate values. In contrast, label Smoothing (Silva & Oliveira, 2021; Liu et al., 2022a) require only modifying the training data. Instead of one-hot encoding $\mathbf{Y}$ in to $[0, 1]$, a small positive random variable $\epsilon$ is introduced, which converts the targets into $[\epsilon, 1 - \epsilon]$, discouraging the model from producing overly confident logit outputs. The Focal Loss (Mukhoti et al., 2020) adds a modulating factor to down-weight the loss from well-classified examples. This forces the model to focus more on hard-to-classify pixels, which can implicitly improve calibration by preventing it from becoming overly certain about easy examples.

Ultimately, relying on raw model outputs as probabilities typically yields sample-level spatial incoherence. Because modern neural networks are often uncalibrated by default, they require post-hoc calibration. Crucially, resolving spatial incoherence does not preclude good calibration, i.e., spatially correlated models can still be well calibrated. Conversely, calibration alone cannot impose spatial structure. Well-calibrated models may remain spatially incoherent. This asymmetry motivates placing greater emphasis on modeling spatial coherence rather than calibration alone.

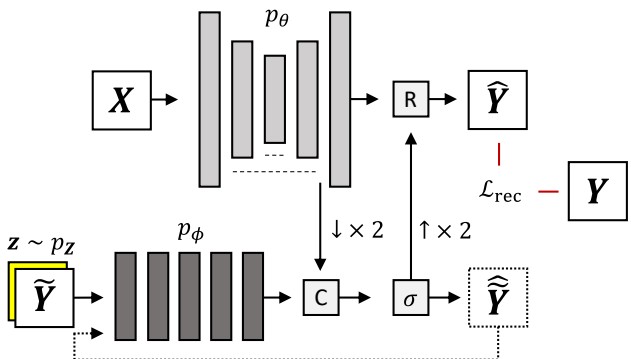

Figure 5: Illustration of the PixelCNN-based PixelSeg (Zhang et al., 2022c). Parameter 'C' indicates the concatenation module, 'R' the resampling module and $\sigma$ the softmax activation. Dotted elements appear during test-time sampling.

### 4.1.2 Spatial correlation

Introducing spatial correlation can be achieved through an autoregressive approach. For instance, we can rephrase Equation (7) to

$$p(Y|X) = \prod_i^{K \times D} p(Y_i|Y_1, ..., Y_{i-1}, X), \tag{10}$$

where pixel $Y_i$ is predicted based on the preceding pixels. A popular implementation of this formulation is known as the PixelCNN (Van Den Oord et al., 2016). For dense predictions, Zhang et al. (2022c) propose PixelSeg, which predicts a downsampled segmentation mask $\tilde{\mathbf{y}}$ with $p_\phi(\tilde{\mathbf{y}}|\mathbf{x})$, and fuse this with a conventional CNN to predict the full-resolution mask with $p_\theta(\mathbf{y}|\tilde{\mathbf{y}}, \mathbf{x})$. The two masks are fused through a resampling module, containing a series of specific transformations to improve quality and diversity of the samples (Figure 5). Notably, the recursive sampling process also enables completion/inpainting of user-given inputs.

Monteiro et al. (2020) propose the Stochastic Segmentation Network (SSN), which models the output logits as a multivariate normal distribution, parameterized by the neural networks $f_\theta^\mu$ and $f_\theta^\Sigma$. Given the output features $f_\theta(\mathbf{x}) = \mathbf{a}$ of a deterministic model, we can denote the logits distribution as

$$p(\mathbf{a}|\mathbf{x}, \boldsymbol{\theta}) = \mathcal{N}(\mathbf{a}; \boldsymbol{\mu} = f_\theta^\mu(\mathbf{x}), \boldsymbol{\Sigma} = f_\theta^\Sigma(\mathbf{x})), \tag{11}$$

where the covariance matrix has a low-rank structure $\boldsymbol{\Sigma} = \mathbf{P}\mathbf{P}^T + \boldsymbol{\Lambda}$, with $\mathbf{P}$ having dimensionality $((K \times D) \times R)$, with $R$ being a hyperparameter that controls the parameterization rank and $\boldsymbol{\Lambda}$ representing a diagonal matrix. The low-rank assumption results in a more structured distribution, while retaining reasonable efficiency. Monte Carlo sampling is used to generate predictions, which are then mapped to categorical values using the SoftMax activation. SSNs can theoretically be augmented to any pretrained CNNs as an additional layer (see Figure 6).

Figure 6: Depiction of a Stochastic Segmentation Network (Monteiro et al., 2020). Here, the covariance of the likelihood distribution is explicitly modeled through a low-rank approximation.

**Key takeaways**: Calibrating standard segmentation models can produce valid confidence values. However, it still preserves the incorrect pixel-independence assumption and requires a separate validation set. A possible solution is to model the correlation directly in pixel space through autoregressive modeling or a low-rank approximation of the output distribution.

## 4.2 Latent-level sampling

The limitation of modeling intricate, complex distributions directly at the output level can be mitigated by using generative models, which often rely on simpler, lower-dimensional latent variables $\mathbf{Z} \sim p_Z$ with $\mathcal{Z} \in \mathbb{R}^d$, to instead learn the approximation through

$$p_{\boldsymbol{\theta},\boldsymbol{\psi}}(\mathbf{Y}|\mathbf{X}) = \int p_{\boldsymbol{\theta}}(\mathbf{Y}|\mathbf{z}, \mathbf{X}) p_{\boldsymbol{\psi}}(\mathbf{z}|\mathbf{X}) \mathrm{d}\mathbf{z}, \tag{12}$$

with parameters $\boldsymbol{\theta}, \boldsymbol{\psi}$. As such, the spatial correlation is induced through mapping the latent variables to segmentation masks. On a related note, an unconditional prior $p(\mathbf{z})$ can also be employed. However, conditioning the latent density, i.e. $p_{\boldsymbol{\psi}}(\mathbf{z}|\mathbf{X})$, is usually preferred for smooth optimization trajectories (Zheng et al., 2022). The key overarching contribution of latent-level sampling methods lies in the conditioning of generative models on the input image. This section briefly introduces Generative Adversarial Networks (Section 4.2.1), Variational Autoencoders (Section 4.2.2), and Denoising Diffusion Probabilistic Models (Section 4.2.3), and outlines how these architectures are commonly adapted for probabilistic segmentation.

### 4.2.1 Generative Adversarial Networks (GANs)

A straightforward approach is to simply learn the marginalization in Equation (12) through sampling from an unconditional prior density, $p_Z$, and mapping this to segmentation $\mathbf{Y}$ through a *generator* $G_{\boldsymbol{\phi}} : \mathcal{X} \times \mathcal{Z} \to \mathcal{Y}$. Goodfellow et al. (2014) show that this approach can be notably enhanced through the incorporation of a discriminative function (the *discriminator*), denoted as $D_{\boldsymbol{\psi}} : \mathbb{R}^{C \times D} \to [0, 1]$. In this way, $G_{\boldsymbol{\phi}}$ learns to reconstruct realistic images using the discriminative capabilities of $D_{\boldsymbol{\psi}}$, making sufficient guidance from $D_{\boldsymbol{\psi}}$ to $G_{\boldsymbol{\phi}}$ imperative. We can denote the cost of $G_{\boldsymbol{\phi}}$ in the GAN as the negative cost of $D_{\boldsymbol{\psi}}$ as

$$J_{G_{\boldsymbol{\phi}}} = -J_{D_{\boldsymbol{\psi}}} = \mathbb{E}_{p_{\mathcal{D}}}[\log D_{\boldsymbol{\psi}}(\mathbf{y})] - \mathbb{E}_{p_{\mathbf{z}}}\mathbb{E}_{p_{\mathcal{D}}}[\log(1 - D_{\boldsymbol{\psi}}(G_{\boldsymbol{\phi}}(\mathbf{z}, \mathbf{x})))]. \tag{13}$$

Conditional GANs have been used for semantic segmentation in early literature (Isola et al., 2017). However, Kassapis et al. (2021) explicitly contextualized the architecture within uncertainty quantification, using their proposed Calibrated Adversarial Refinement (CAR) network (see Figure 7). The calibration network, $F_{\boldsymbol{\theta}} : \mathbb{R}^{C \times D} \to \mathbb{R}^{K \times D}$, initially provides a SoftMax activated prediction as $F_{\boldsymbol{\theta}}(\mathbf{x}) = c$, with (cross-entropy) reconstruction loss

$$\mathcal{L}_{\mathrm{rec}} = -\mathbb{E}_{p_{\mathcal{D}}}[\log p_{\boldsymbol{\theta}}(\mathbf{c}|\mathbf{x})]. \tag{14}$$

Then, the conditional refinement network $G_{\boldsymbol{\theta}}$ uses $\mathbf{c}$ together with input image $\mathbf{x}$ and latent samples $\mathbf{z}_i \sim p_Z$ injected at multiple decomposition scales $i$, to predict various segmentation maps. Furthermore, the refinement network is subject to the adversarial objective

$$\mathcal{L}_{\mathrm{adv}} = -\mathbb{E}_{p_{\mathcal{D}}}\mathbb{E}_{p_{\mathbf{z}}}[\log D_{\boldsymbol{\psi}}(G_{\boldsymbol{\phi}}(F_{\boldsymbol{\theta}}(\mathbf{x}), \mathbf{z}), \mathbf{x})], \tag{15}$$

which is argued to elicit superior structural qualities compared to relying solely on cross-entropy loss. At the same time, the discriminator opposes the optimization with

$$\mathcal{L}_D = -\mathbb{E}_{p_{\mathbf{z}}}\mathbb{E}_{p_{\mathcal{D}}}[1 - \log D_{\boldsymbol{\psi}}(G_{\boldsymbol{\phi}}(F_{\boldsymbol{\theta}}(\mathbf{x}), \mathbf{z}), \mathbf{x})] - \mathbb{E}_{p_{\mathcal{D}}}[\log D_{\boldsymbol{\psi}}(\mathbf{y})]. \tag{16}$$

Finally, the average of the $N$ segmentation maps generated from $G_{\boldsymbol{\phi}}$ are compared against the initial prediction of $F_{\boldsymbol{\theta}}$ through the calibration loss, which is the analytical KL-divergence between the two categorical densities, denoted by

$$\mathcal{L}_{\mathrm{cal}} = \mathbb{E}_{p_{\mathcal{D}}} \mathrm{KL}[p_{\boldsymbol{\phi}}(\mathbf{y}|\mathbf{c}, \mathbf{x}) \,\|\, p_{\boldsymbol{\theta}}(\mathbf{c}|\mathbf{x})]. \tag{17}$$

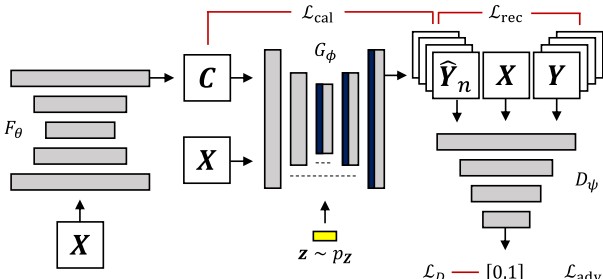

Figure 7: Diagram of the Calibrated Adversarial Refinement network (Kassapis et al., 2021), based on the Generative Adversarial Network (Goodfellow et al., 2014) with additional mechanisms and loss terms.

In this way, the generator loss can be defined as

$$\mathcal{L}_G = \mathcal{L}_{\text{adv}} + \lambda \cdot \mathcal{L}_{\text{cal}}, \tag{18}$$

with hyperparameter $\lambda \geq 0$. The purpose of the calibration network is argued to be threefold. Namely, it (1) sets a calibration target for $\mathcal{L}_{\text{cal}}$, (2) provides an alternate representation of $\mathbf{X}$ to $G_\phi$, and (3) allows for sample-free uncertainty quantification. The refinement network can be seen as modeling the spatial dependency across the pixels, which enables sampling coherent segmentation maps.

### 4.2.2 Variational Autoencoders (VAEs)

Techniques such as GANs rely on implicit distributions and are void of any notion of data likelihoods. An alternative approach estimates the Bayesian posterior w.r.t. the latent variables, $p(\mathbf{Z}|\mathbf{Y}, \mathbf{X})$, with an approximation $q_\theta(\mathbf{Z}|\mathbf{Y}, \mathbf{X})$, obtained by maximizing the conditional Evidence Lower Bound (ELBO)

$$\begin{aligned}
\log p(\mathbf{Y}|\mathbf{X}) &= \log \int q_\theta(\mathbf{z}|\mathbf{Y}, \mathbf{X}) \frac{p(\mathbf{z}, \mathbf{Y}|\mathbf{X})}{q_\theta(\mathbf{z}|\mathbf{Y}, \mathbf{X})} \mathrm{d}\mathbf{z} \\
&\geq \int q_\theta(\mathbf{z}|\mathbf{Y}, \mathbf{X}) \log \frac{p(\mathbf{z}, \mathbf{Y}|\mathbf{X})}{q_\theta(\mathbf{z}|\mathbf{Y}, \mathbf{X})} \mathrm{d}\mathbf{z} \\
&= \mathbb{E}_{q_\theta(\mathbf{Z}|\mathbf{Y}, \mathbf{X})} [\log p_\phi(\mathbf{Y}|\mathbf{z}, \mathbf{X})] - \mathrm{KL}[q_\theta(\mathbf{z}|\mathbf{Y}, \mathbf{X})||q_\psi(\mathbf{z}|\mathbf{X})],
\end{aligned} \tag{19}$$

where Jensen's inequality justifies moving the logarithm inside the integral. The first term in Equation (19) represents the reconstruction cost of the decoder, subject to the latent code $\mathbf{Z}$ and input image $\mathbf{X}$. The second term is the Kullback-Leibler (KL) divergence between the approximate posterior and prior density. As a consequence of the mean-field approximation, all involved densities are modeled by axis-aligned gaussian densities and amortized through neural networks, parameterized by $\phi$, $\theta$ and $\psi$. The predictive distribution after observing dataset $\mathcal{D}$ is then obtained as

$$p(\mathbf{Y}|\mathbf{x}^*) = \int p_\phi(\mathbf{Y}|\mathbf{z}, \mathbf{x}^*) q_\theta(\mathbf{z}|\mathbf{x}^*) \mathrm{d}\mathbf{z}. \tag{20}$$

The implementation of the conditional ELBO in Equation (19) can be achieved through the well known VAE architecture (Kingma & Welling, 2013). Furthermore, some additional design choices lead to the segmentation-specific Probabilistic U-Net (PU-Net) (Kohl et al., 2019b). Firstly, the latent variable $\mathbf{Z}$ is only introduced at the final layers of a U-Net conditioned on the images. The latent vector is up-scaled through tiling and then concatenated with the feature maps of the penultimate layer, which is followed by a sequence of $1 \times 1$ convolution for classification. When involving conditional latent variables in this manner, it is expected that most of the semantic feature extraction and delineation are performed in the U-Net, while the variability in the latent samples is almost exclusively related to the segmentation variability. Therefore, relatively smaller values of $d$ are feasible than what is commonly used in conventional image generation tasks. Similar to related research on the VAE (Zhao et al., 2017; Bousquet et al., 2017; Higgins et al., 2017; Van Den Oord et al., 2017; Rezende & Mohamed, 2015), much work has been dedicated to improving the PU-Net.

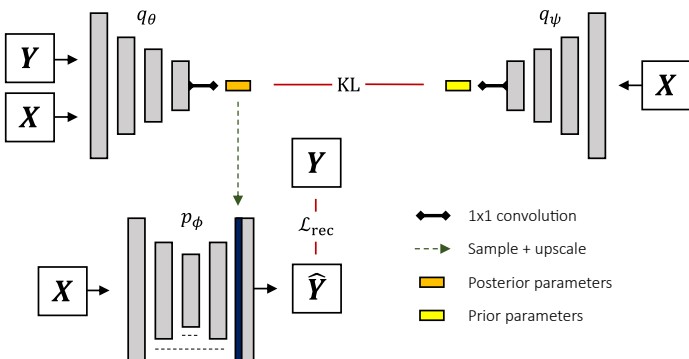

Figure 8: Depiction of the Probabilistic U-Net (Kohl et al., 2019b) based on a conditional Variational Autoencoder (Kingma & Welling, 2013). The latent samples are inserted at the final stages of a U-Net through a tiling operation.

**Density reparameterization**  Augmenting a Normalizing Flow (NF) to the posterior density of a VAE is a commonly used tactic to improve its expressiveness (Rezende & Mohamed, 2015). This phenomenon has also been successfully applied to VAE-like models such as the PU-Net (Valiuddin et al., 2021; Selvan et al., 2020). NFs are a class of generative models that utilize $k$ consecutive bijective transformations $f_k : \mathbb{R}^D \to \mathbb{R}^D$ as $f = f_K \circ \ldots \circ f_k \circ \ldots \circ f_1$, to express exact log-likelihoods of arbitrarily complex approximations of the posterior $p(\mathbf{z}|\mathbf{x})$, where symbol $\circ$ indicates functional composition. The approximated posteriors in turn induce an approximate of $p(\mathbf{x}|\mathbf{z}) = p(\mathbf{z}|\mathbf{x})\frac{p(\mathbf{x})}{p(\mathbf{z})}$ and are often denoted as $\log p(\mathbf{x})$ for simplicity. The fractional term accounts for the change in probability space and corresponds to the log-Jacobian determinant, as given by the Change of Variables theorem. Thus, we can denote

$$\log p(\mathbf{x}) = \log p_{\mathbf{Z}}(\mathbf{z}_0) - \sum_{k=1}^{K} \log \left| \det \frac{\mathrm{d}f_k(\mathbf{z}_{k-1})}{\mathrm{d}\mathbf{z}_{k-1}} \right|, \tag{21}$$

where $\mathbf{z}_k$ and $\mathbf{z}_{k-1}$ are variables from intermediate densities and $\mathbf{z}_0 = \mathbf{f}^{-1}(\mathbf{x})$. Equation (21) can be substituted in the conditional ELBO objective in Equation (19) to obtain

$$\begin{aligned} p(\mathbf{Y}|\mathbf{X}) \geq\ & \mathbb{E}_{q_\theta(\mathbf{Z}|\mathbf{Y},\mathbf{X})} \big[\log p_\phi(\mathbf{Y}|\mathbf{z},\mathbf{X})\big] \\ & - \mathrm{KL}\big[\, q_\theta(\mathbf{z}_0|\mathbf{Y},\mathbf{X}) || q_\psi(\mathbf{z}_k|\mathbf{X})\,\big] - \mathbb{E}_{q_\theta(\mathbf{z}_0|\mathbf{Y},\mathbf{X})} \left[ \sum_{k=1}^{K} \log \left| \det \frac{\mathrm{d}f_k(\mathbf{z}_{k-1})}{\mathrm{d}\mathbf{z}_{k-1}} \right| \right], \end{aligned} \tag{23}$$

where the objective consists of a reconstruction term, sample-based KL-divergence and a likelihood correction term for the change in probability density induced by the NF. Bhat et al. (2023; 2022a) compare this approach with other parameterizations of the latent space, including a mixture of Gaussians and low-rank approximation of the full covariance matrix. Valiuddin et al. (2024b) show that the latent space can converge to contain non-informative latent dimensions, undermining the capabilities of the latent-variable approach, generally referred to as mode or posterior collapse (Chen et al., 2016; Zhao et al., 2017). Their proposition considers the alternative formulation of the ELBO, specified as

$$\begin{aligned} \log p(\mathbf{Y}|\mathbf{X}) \geq\ & \mathbb{E}_{q_\theta(\mathbf{Z}|\mathbf{Y},\mathbf{X})} \left[\log p_\phi(\mathbf{Y}|\mathbf{X},\mathbf{z})\right] \\ & - \mathrm{KL}[q_\theta(\mathbf{z}|\mathbf{X}) \,||\, q_\psi(\mathbf{z}|\mathbf{X})] - I(\mathbf{Y},\mathbf{Z}|\mathbf{X}). \end{aligned} \tag{23}$$

The proposed novel objective maximizes the contribution of the (expected) mutual information between latent and output variables, i.e. the aleatoric uncertainty term of the predictive entropy defined later in Equation (42). This enables the introduction of the updated objective

$$\begin{aligned} \mathcal{L} =\ & -\mathbb{E}_{q_\theta(\mathbf{Z}|\mathbf{Y},\mathbf{X})}[\log p_\phi(\mathbf{Y}|\mathbf{X},\mathbf{z})] \\ & + \alpha \cdot \mathrm{KL}[q_\theta(\mathbf{z}|\mathbf{Y},\mathbf{X}) \,||\, p_\psi(\mathbf{z}|\mathbf{X})] + \beta \cdot \mathrm{S}_\epsilon[q_\theta(\mathbf{z}|\mathbf{X}) \,||\, p_\psi(\mathbf{z}|\mathbf{X})], \end{aligned} \tag{24}$$

with $S_\epsilon$ being the Sinkhorn divergence (Cuturi, 2013) and $\alpha, \beta$ denoting hyperparameters. This adaptation results in a more uniform latent space leading to increased model performance. Also, modeling the ELBO of the joint density has been explored (Zhang et al., 2022b). This formulation results in an additional reconstruction term and forces the latent variables to be more congruent with the data. Furthermore, constraining the latent space to be discrete has resulted in some improvements, where it is hypothesized that this partially addresses the model collapse phenomenon (Qiu & Lui, 2020).

**Multi-scale approach** Learning latent features over several hierarchical scales can provide expressive densities and interpretable features across various abstraction levels (Sønderby et al., 2016; Kingma et al., 2016; Klushyn et al., 2019; Gregor et al., 2015; Ranganath et al., 2016). Such models are commonly categorized under hierarchical VAE (HVAE) modeling. Often, an additional Markov assumption of length $T$ is placed on the posterior as

$$q_\theta(\mathbf{Z}_{1:T}|\mathbf{Y}, \mathbf{X}) = q_\theta(\mathbf{Z}_1|\mathbf{Y}, \mathbf{X}) \prod_{t=2}^{T} q_\theta(\mathbf{Z}_t|\mathbf{Z}_{t-1}, \mathbf{X}). \tag{25}$$

Consequently, the conditional ELBO is denoted as

$$\begin{aligned}
p(\mathbf{Y}|\mathbf{X}) \geq\ & \mathbb{E}_{q_\theta(\mathbf{Z}|\mathbf{Y}, \mathbf{X})}[\log p_\phi(\mathbf{Y}|\mathbf{z}, \mathbf{X})] \\
& - \sum_{t=2}^{T} \mathrm{KL}[q_\theta(\mathbf{z}_t|\mathbf{Y}, \mathbf{X}, \mathbf{z}_{1:t-1})||q_\psi(\mathbf{z}_t|\mathbf{z}_{1:t-1})] - \mathrm{KL}[q_\theta(\mathbf{z}_1|\mathbf{Y}, \mathbf{X})||q_\psi(\mathbf{z}_1|\mathbf{X})].
\end{aligned} \tag{26}$$

This objective is implemented in the Hierarchical PU-Net (Kohl et al., 2019a) (HPU-Net, depicted in Figure 9), which learns a latent representation at multiple decomposition levels of the U-Net. Residual connections in the convolutional layers are necessary to prevent degeneracy of uninformative latent variables with the KL divergence rapidly approaching zero. For similar reasons, the Generalized ELBO with Constrained Optimization (GECO) objective is employed, which extends Equation (26) to

$$\begin{aligned}
\mathcal{L}_{\mathrm{GECO}} =\ & \lambda \cdot ||\mathbb{E}_{q_\theta(\mathbf{Z}|\mathbf{Y}, \mathbf{X})}[\log p_\phi(\mathbf{Y}|\mathbf{z}, \mathbf{X})] - \kappa||_1 \\
& - \sum_{t=2}^{T} \mathrm{KL}[q_\theta(\mathbf{z}_t|\mathbf{Y}, \mathbf{X}, \mathbf{z}_{1:t-1})\,||\,q_\psi(\mathbf{z}_t|\mathbf{z}_{1:t-1})] - \mathrm{KL}[q_\theta(\mathbf{z}_1|\mathbf{Y}, \mathbf{X})\,||\,q_\psi(\mathbf{z}_1|\mathbf{X})].
\end{aligned} \tag{27}$$

Hyperparameter $\lambda$ is the Lagrange multiplier update through the Exponential Moving Average of the reconstruction, which is constrained to reach target value $\kappa$, empirically set beforehand to an appropriate value. Finally, online negative hard mining is used to only backpropagate 2% of the worst performing pixels, which are stochastically picked with the Gumbel-SoftMax trick (Jang et al., 2016; Huijben et al., 2022). Furthermore, PHiSeg (Baumgartner et al., 2019) takes a similar approach to the HPU-Net. However, PHiSeg places residual connections between latent vectors across decompositions rather than in the convolutional layers. Furthermore, *deep supervision* is placed at each decomposition scale to enforce disentanglement between the latent variables.

**Extension to 3D** Early methods for uncertainty quantification in medical imaging primarily utilized 2D slices from three-dimensional (3D) datasets, leading to a potential loss of critical spatial information and subtle nuances often necessary for accurate delineation. This limitation has spurred research into 3D probabilistic segmentation techniques with ELBO-based models, aiming to preserve the integrity of entire 3D structures. Initial works (Chotzoglou & Kainz, 2019; Long et al., 2021a) demonstrate that the PU-Net can be adapted by replacing all 2D operations with their 3D variants. Crucially, the fusion of the latent sample with 3D extracted features is achieved through a 3D tiling operation. Viviers et al. (2023c) additionally augment a Normalizing Flow to the posterior density for improved expressiveness. Further enhancements to the architecture include the implementation of the 3D HPU-Net (Saha et al., 2020), an updated feature network incorporating the attention mechanisms, a nested decoder, and different reconstruction loss components tailored to specific applications (Saha et al., 2021a).

**Conditioning on annotator** It can be relevant to model the annotators independently in cases with consistent annotator-segmentation pairs in the dataset. This can be achieved by conditioning the learned densities on the annotator itself (Gao et al., 2022; Schmidt et al., 2023). For example, features of a U-Net can be combined with samples from annotator-specific gaussian-distributed posteriors (Schmidt et al.,

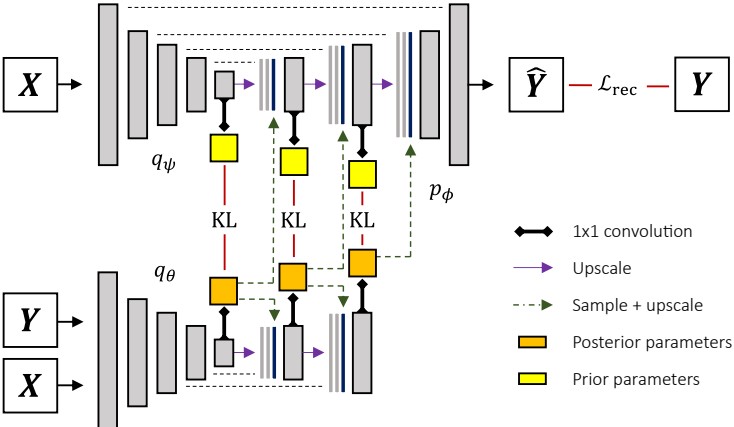

Figure 9: Hierarchical Probabilistic U-Net (Kohl et al., 2019a) based on a hierarchical Variational Autoencoder (Kingma & Welling, 2013; Sønderby et al., 2016; Klushyn et al., 2019). Instead of a single latent code, multiple decomposition scales encode the segmentation variability.

2023). Considering the approach from Gao et al. (2022), generating a segmentation mask is achieved by first sampling an annotator from a categorical prior distribution $\mathcal{C}(\pi_k(\mathbf{x}))$, governed by the image conditional parameters $\pi_k(\mathbf{x})$ for the $k$-th annotator. Then, samples are taken from its corresponding prior density as $\mathbf{z}_k \sim p_k(\mathbf{z}_k)$ to reconstruct a segmentation through an image-conditional decoder $y = F(\mathbf{x}, \mathbf{z}_k)$. The parameters $\pi_k(\mathbf{z}_k)$ can also be used to weigh the corresponding predictions to express the uncertainty in the prediction ensemble. Additionally, consistency between the model and ground-truth distribution is enforced through an optimal transport loss between the set of predictions and labels.

### 4.2.3 Denoising Diffusion Probabilistic Models (DDPMs)

Recent developments in generative modeling have resulted in a family of models known as Denoising Diffusion Probabilistic Models (Ho et al., 2020; Song et al., 2022; 2020). Such models are especially renowned for their expressive power by being able to encapsulate large and diverse datasets. While several perspectives can be used to introduce the DDPMs, we build upon the earlier discussed HVAE (Section 4.2.2) to maintain cohesiveness with the overall manuscript. Specifically, we describe three additional modifications to the HVAE (Luo, 2022). Firstly, the latent dimensionality is set equal to the spatial data dimensions, i.e. $d = D$. As a consequence, redundant notation of $\mathbf{Z}$ is removed and $\mathbf{Y}$ is instead subscripted with $t \in \{1, ..., T\}$, indicating the encoding depth, where $\mathbf{Y}_0$ is the initial segmentation mask. Secondly, the encoding process (or *forward process*) is predefined as a linear Gaussian model such that

$$q(\mathbf{Y}_T|\mathbf{Y}_0) = p(\mathbf{Y}_0) \prod_{t=1}^{T} q(\mathbf{Y}_t|\mathbf{Y}_{t-1}), \tag{28}$$

with $q(\mathbf{y}_t|\mathbf{Y}_{t-1}) = \mathcal{N}(\mathbf{y}_t; \boldsymbol{\mu} = \sqrt{\alpha_t}\,\mathbf{Y}_{t-1}, \boldsymbol{\Sigma} = (1 - \alpha_t) \cdot \mathbf{I})$, and noise schedule $\boldsymbol{\alpha} = \{\alpha_t\}_{t=1}^{T}$. Then, the decoding or *reverse process* can be learned through $p_\phi(\mathbf{Y}_{t-1}|\mathbf{Y}_t, \mathbf{x})$. The ELBO for this objective is defined as

$$p(\mathbf{Y}|\mathbf{X}) \geq \mathbb{E}_{q(\mathbf{Y}_1|\mathbf{Y}_0)}\big[\log p_\phi(\mathbf{Y}_0|\mathbf{y}_1, \mathbf{X})\big]$$
$$+ \sum_{t=2}^{T} \mathbb{E}_{q(\mathbf{Y}_t|\mathbf{Y}_0)}\big[\mathrm{KL}[q(\mathbf{y}_{t-1}|\mathbf{y}_t)\,||\,p_\phi(\mathbf{y}_{t-1}|\mathbf{y}_t, \mathbf{X})]\big] + \underbrace{\mathbb{E}_{q(\mathbf{Y}_T|\mathbf{Y}_0)}\left[\log \frac{p(\mathbf{y}_T)}{q(\mathbf{y}_T|\mathbf{Y}_0)}\right]}_{\approx 0}. \tag{29}$$

As denoted, the regularization term is ignored, since it is assumed that a sufficient amount of steps $T$ are taken such that $q(\mathbf{y}_T|\mathbf{y}_0)$ is approximately normally distributed. With the reparameterization trick (Kingma & Welling, 2013), the forward process is governed by random variable $\boldsymbol{\epsilon} \sim \mathcal{N}(0, 1)$. As such, the KL divergence in the second term can be interpreted as predicting $\mathbf{Y}_0$, the source noise $\boldsymbol{\epsilon}$ or the score $\nabla_{\mathbf{Y}_t} \log q(\mathbf{y}_t)$ (gradient

of the data log-likelihood) from $\mathbf{Y}_t$, depending on the parameterization. This is in almost all instances approximated with U-Net variants (Ronneberger et al., 2015; Song et al., 2020).

The proposed methodologies for segmentation vary in the conditioning of the reverse process on the input image. For instance, Wolleb et al. (2021) concatenate the input image with the noised segmentation mask. Wu et al. (2023b) insert encoded image features to the U-Net bottleneck. Additionally, information on predictions $Y_t$ at a time step $t$ is provided in the intermediate layers of the conditioning encoder. This is performed by applying the Fast Fourier Transform (FFT) on the U-Net encoding, followed by a learnable attention map and the inverse FFT. The procedure of applying attention on the spectral domain of the U-Net encoding has also been implemented with transformers in follow-up work (Wu et al., 2023a). Amit et al. (2021) also encode both current time step and input image, but combine the extracted features by simple summation prior to inferring it through the U-Net. It has also been proposed to model Bernoulli instead of Gaussian noise (Chen et al., 2023a; Zbinden et al., 2023; Rahman et al., 2023; Bogensperger et al., 2023).

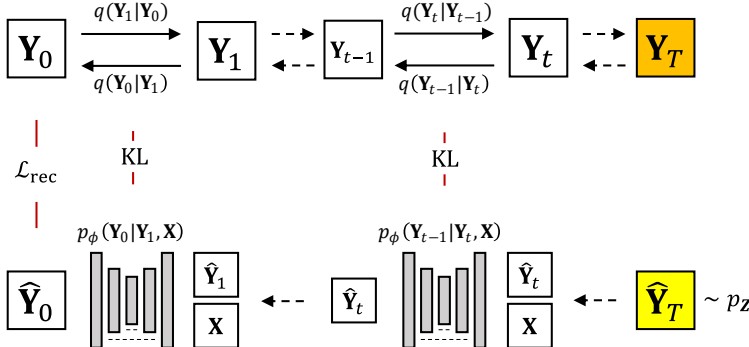

Figure 10: Illustration of conditional Diffusion Probabilistic Models (Ho et al., 2020). The model learns to remove noise that has been gradually added to the input image.

> **Key takeaways**: There is a large focus on improving amortized inference in VAE-based models (richer posteriors, hierarchies, refined objectives). The GAN-based CAR model has received limited follow-up. More contemporary work incline towards using DDPMs, where innovations largely center on input-conditioning strategies and employing categorical distributions in the noising process.

## 5    Parameter modeling

A significant advantage of this parameter-sampling approach, when compared to modeling the output distribution directly, is that it naturally induces spatial coherence across the output segmentations. As a result, existing methods are often simple extensions of uncertainty modeling in conventional classification, rather than leveraging techniques tailored for segmentation tasks. As established in Section 3, the posterior distribution from Equation (1) is used to generate a predictive distribution via Equation (2), suggesting parameter distribution modeling as an alternative to the modeling in feature space. We refer to any neural network that approximates the Bayesian posterior over model parameters as a Bayesian Neural Network (BNN). However, evaluating the true Bayesian posterior is impeded by the intractability of the data likelihood in the denominator. This section discusses the methods used in image segmentation for approximating the parameter distribution. The most commonly used approach is Variational Inference (VI) over model parameters (Section 5.1). Laplace Approximations are examined in Section 5.2. Test-Time Augmentation (TTA) is covered in Section 5.3 (we justify this non-trivial choice later in Section 7.1).

## 5.1 Variational Inference

Consider a simpler, tractable density $q(\boldsymbol{\theta}|\boldsymbol{\eta})$, parameterized by $\boldsymbol{\eta}$, to approximate posterior $p(\boldsymbol{\theta}|\mathbf{y}, \mathbf{x})$. Variational Inference (VI) w.r.t. to the parameters can be employed, by minimizing the Kullback-Leibler (KL) divergence between the true and approximated Bayesian posterior as

$$
\begin{aligned}
\boldsymbol{\eta}^* &= \arg\min_{\boldsymbol{\theta}} \mathrm{KL}\left[\, q(\boldsymbol{\theta}|\boldsymbol{\eta}) \,||\, p(\boldsymbol{\theta}|\mathcal{D}) \,\right] \\
&= \arg\min_{\boldsymbol{\theta}} \int q(\boldsymbol{\theta}|\boldsymbol{\eta}) \log \frac{q(\boldsymbol{\theta}|\boldsymbol{\eta})}{p(\mathbf{Y}|\mathbf{x}, \boldsymbol{\theta})p(\mathbf{x}, \boldsymbol{\theta})} d\boldsymbol{\theta} \\
&= \arg\min_{\boldsymbol{\theta}} \mathrm{KL}\left[q(\boldsymbol{\theta}|\boldsymbol{\eta}) \,||\, p(\boldsymbol{\theta})\right] - \mathbb{E}_{q(\boldsymbol{\theta}|\boldsymbol{\eta})}[\, \log p(\mathbf{y}|\mathbf{x}, \boldsymbol{\theta}) \,],
\end{aligned}
\tag{30}
$$

where the parameter-independent terms are constant and therefore excluded from notation. In the case of a deterministic encoder, i.e. $q(\boldsymbol{\theta}|\boldsymbol{\eta}) = \delta(\boldsymbol{\theta} - \boldsymbol{\theta}^*)$, the formulation collapses to the MAP estimate in Equation (5). A popular choice for the approximated variational posterior is the gaussian distribution, i.e., a mean $\mu$ and covariance $\sigma$ parameter for each element of the convolutional kernel, usually with zero-mean unitary gaussian prior densities. However, the priors can be also learned through Empirical Bayes (Bishop, 1995). Furthermore, backpropagation is possible with the *reparameterization trick* (Kingma & Welling, 2013) and within this context, the procedure is referred to as Bayes by Backprop (BBB) (Blundell et al., 2015) and has later been improved with the Local Reparameterization trick (Kingma et al., 2015). In addition to approximating the posterior, sampling from it yields multiple parameter permutations, effectively enriching the model's hypothesis space, a phenomenon also referred to as model combination (Minka, 2000; Clarke, 2003; Lakshminarayanan, 2016). In most cases, VI is not performed explicitly. Instead, simpler techniques are employed, with the resulting parameter permutations hypothesized to serve as a proxy. We discuss two such methods: Monte Carlo Dropout (Section 5.1.1) and Ensembling (Section 5.1.2). See Figure 11 for an illustration how each method approximates the convolutional kernel distribution. All of these methods have been widely applied to segmentation. However, unlike feature-level distribution modeling, their implementations are often more generic and not specifically tailored to segmentation tasks. Therefore, we provide only a brief overview here, followed by a more in-depth discussion in later sections related to various applications.

### 5.1.1 Monte Carlo Dropout

Dropout is a common technique used to regularize neural networks by randomly "switching off" nodes of the neural network. Using Dropout can also be interpreted as a first-order equivalent $L_2$ regularization with additionally transforming the input with the inverse diagonal Fisher information matrix (Wager et al., 2013). Furthermore, with *Monte-Carlo Dropout* (MC Dropout), the random node switching is continued during test time, effectively sampling new sets of parameters, mimicking samples from an implicit parameter distribution $q(\tilde{\boldsymbol{\theta}}|\boldsymbol{\theta}, p)$, defined as

$$
\mathbf{n} \sim \mathrm{Bernoulli}(p), \tag{31a}
$$

$$
\tilde{\boldsymbol{\theta}} = \boldsymbol{\theta} \odot \mathbf{n}, \tag{31b}
$$

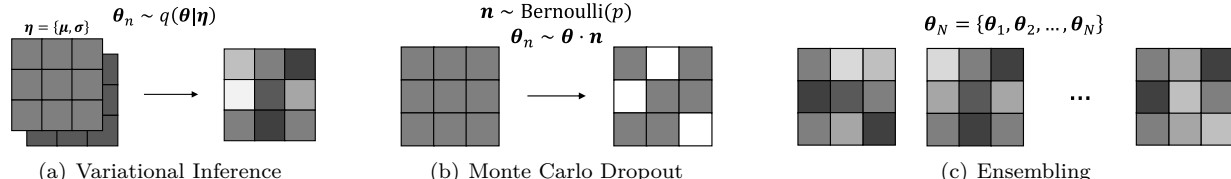

(a) Variational Inference      (b) Monte Carlo Dropout      (c) Ensembling

Figure 11: Three techniques for sampling convolutional kernels, with grayscale values indicating kernel weight magnitudes. These include: (a) Sampling an gaussian approximation parameterized by a mean and standard deviation kernel. (b) Applying dropout at inference time. (c) Training $N$ independently trained models, providing a discrete set of parameter configurations.

with probability $p$ and vector $\mathbf{n}$ operating element-wise on the parameters. It has been shown that MC Dropout can be interpreted as approximate VI in a Deep Gaussian Process (Gal & Ghahramani, 2016). In this manner, such a method is able to provide multimodal estimates of the uncertainty. The dropout rate is usually optimized through grid search or simply set to $p = 0.5$. The relationship between $p$ and the magnitude of the model weights can also be exploited to probabilistically prune neural networks (Gonzalez-Carabarin et al., 2022). As noted by Gal et al. (2017a), the variance in the model output is primarily governed by the magnitudes of the weights rather than the dropout rate $p$. Hence, they propose to additionally learn $p$ using gradient-based methods, known as Concrete Dropout, to increase the influence of $p$. As the name suggests, a continuous approximation to the discrete distribution is used, known as the Concrete distribution (Maddison et al., 2016; Jang et al., 2016), to enable path-wise derivatives through $p$. Also, a generalization of MC dropout has been proposed, where the weights of the network, rather than its hidden output, are set to zero (Mobiny et al., 2021).

### 5.1.2 Ensembling

As mentioned earlier, MC dropout effectively optimizes over a set of sparse neural networks. From this perspective, explicitly ensembling multiple models can also be considered as an approximation of VI (Lakshminarayanan et al., 2017). To this end, we define the set of functions $\boldsymbol{f} = \{f_{\boldsymbol{\theta}}^n\}_{n=1}^M$, with $M$ representing the number of models in the ensemble. Then, it is relatively simple to obtain $\boldsymbol{\Theta} = \{\boldsymbol{\theta}_m\}_{n=1}^M$, which can be interpreted as samples from an approximate posterior. Ensembling in only the latter parts of a neural network (typically the decoder) is referred to as M-heads, i.e. the network has multiple outputs. Often, the $M$ obtained parameters are from $M$ separate training sessions. However, it has also been proven effective to ensemble from a single training session, by saving the parameters at multiple stages or training with different weight initializations (Dahal et al., 2020; Xie et al., 2013; Huang et al., 2017). A closely related concept to ensembling is known as Mixture of Experts (MoE), where each model in the ensemble (an 'expert') is trained on specific subsets of the data (Jacobs et al., 1991). In such settings, however, a gating mechanism is usually applied after combining the expert hypotheses.

### 5.2 Laplace approximation

It can be shown that the intractable posterior in Equation (2) can be estimated as

$$
\begin{aligned}
p(\boldsymbol{\theta}|\mathcal{D}) &\approx \frac{(\det \mathbf{H})^{\frac{1}{2}}}{(2\pi)^{\frac{d}{2}}} \exp\left(-\frac{1}{2}(\boldsymbol{\theta} - \boldsymbol{\theta}_{\mathrm{MAP}})^T \mathbf{H}\,(\boldsymbol{\theta} - \boldsymbol{\theta}_{\mathrm{MAP}})\right) \\
&= \mathcal{N}(\boldsymbol{\theta}|\mu = \boldsymbol{\theta}_{\mathrm{MAP}}, \Sigma = \mathbf{H}^{-1}),
\end{aligned}
\tag{32}
$$

with $\boldsymbol{\theta}_{\mathrm{MAP}}$ being the Maximum a Posteriori solution and Hessian matrix $\mathbf{H} = -\nabla^2 \log h(\boldsymbol{\theta})|_{\boldsymbol{\theta}_{\mathrm{MAP}}}$, i.e. the second derivative of the loss function evaluated at $\boldsymbol{\theta}_{\mathrm{MAP}}$. This approach, known as the Laplace Approximation (LA) (Laplace, 1774), has the merit that it can be performed post-hoc on pretrained neural networks. However, estimating the Hessian can become infeasible, as computation scales quadratically with the model parameters. This is usually circumvented by treating neural networks partly probabilistic and/or approximating the Hessian with a more simplified matrix structure (Martens & Grosse, 2015; Botev et al., 2017; Daxberger et al., 2021).

### 5.3 Test-time augmentation

An image $\mathbf{X}$ can be understood as a one-of-many visual representations of the object of interest. For example, systematic noise, translation or rotation result in many realistic variations that approximately retain image semantics. Hence, data augmentation has been used at test time (explaining the name test-time augmentation, or TTA), argued to obtain uncertainty estimates by efficiently exploring the locality of the likelihood function (Ayhan & Berens, 2022). By randomly augmenting input images with invertible transformation $\tilde{\mathbf{x}} = T_\zeta(\mathbf{x})$, with transformation parameters $\zeta$, a prediction is obtained with $\tilde{\mathbf{y}} = f_{\boldsymbol{\theta}}(\tilde{\mathbf{x}})$ and can then be inverted through $\mathbf{y} = T_\zeta^{-1}(\tilde{\mathbf{y}})$. Repeatedly performing this procedure results in a set of segmentation masks.

> **Key takeaways**: Compared with earlier methods, parameter distribution modeling is largely model agnostic and rarely incorporates segmentation-specific architectures. In practice, many implementations reduce to approximate Variational Inference techniques, such as MC Dropout or Ensembling, followed by methods like Test-Time Augmentation and Laplace Approximations. Echoing their origins in classical classification, these concepts apply broadly to other domains.

# 6 Tasks

This section reviews literature on the downstream utilization of uncertainty in segmentation models, classifying the corresponding tasks into four main groups. Furthermore, Table 5 in Section 6.5 explicitly connects the downstream tasks to the specific application domains where they have been successfully employed.

> **Tasks**:
>
> - Estimating **Observer Variability**, the range of plausible segmentation masks (Section 6.1)
> - Reducing labeling costs using **Active Learning** (Section 6.2)
> - Giving the model the ability to self-assess with **Model Introspection** (Section 6.3)
> - Improving **Model Generalization** (Section 6.4)

## 6.1 Observer variability

After observing sufficient data, the variability in the predictive distribution is often considered to be negligible and is therefore omitted. Nevertheless, this assumption becomes excessively strong in ambiguous modalities, where its consequence is often apparent with multiple varying, yet plausible annotations for a single image. Additionally, such annotations can also vary due to differences in expertise and experience of annotators. The phenomenon of inconsistent labels across annotators is known as the *inter*-observer variability, while variations from a single annotator are referred to as the *intra*-observer variability (see Figure 12).

For example, in critical applications like radiation therapy planning, the precise delineation of tumor contours is essential. However, inherent ambiguity in the image acquisition process often leads to significant inter-observer variability. Multiple expert radiologists might annotate the same tumor with different, yet equally plausible, boundaries. In this context, no single annotation can be considered definitively correct, and this irreducible uncertainty in the ground truth must be accounted for to mitigate potentially serious impacts on patient care. Similarly, the perception of an autonomous driving system can suffer from label-level uncertainty. Consider an object partially obscured at a distance. The sensor data may be insufficient to distinguish whether it is a pedestrian about to cross the road or a stationary object like a mailbox due to a genuine ambiguity in the scene itself. The decision-making process must handle this uncertainty, as the consequences of misclassifying a vulnerable road user are severe.

**Annotators are models** To contextualize this phenomenon within the framework of uncertainty quantification, annotators can be regarded as models themselves. For example, consider $K$ separate annotators modeled through parameters $\phi_k$ with $k = 1, 2, ..., K$. For a simple segmentation task, it can be expected that $\text{Var}[p(\phi_k)] \to 0$. In other words, each annotator is consistent in their delineation and the *intra*-observer variability is low. For cases with consensus across experts, i.e. yielding negligible inter-observer variability, the marginal approaches to $\text{Var}[p(\phi)] \to 0$. Asserting these two assumptions, it is valid to simply consider a point estimate of the posterior. Yet, this is rarely the case in many real-life applications and, as such, explicitly modeling the involved distributions becomes necessary.

**Evaluation** For evaluation, a commonly used metric minimizes the squared distance between arbitrary mean embeddings of the ground truth and predicted annotations using the kernel trick (Shawe-Taylor, 2004). This metric is known as Maximum Mean Discrepancy (MMD) or the Generalized Energy Dis-

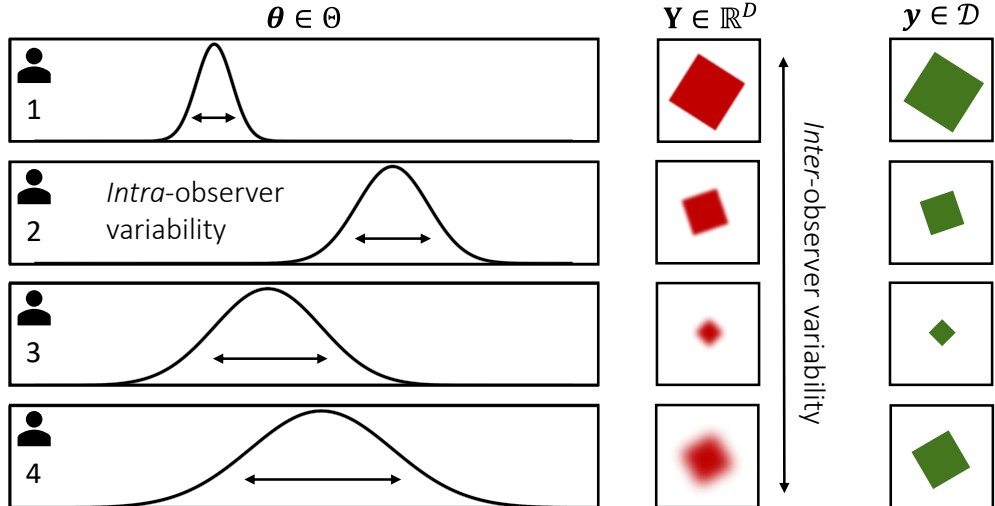

Figure 12: Visualization of the intra-observer variability in parameter space (left) and the inter-observer variability in data space (right).

tance (GED) (Gretton et al., 2012), denoted as

$$\text{GED}(P_\mathbf{Y}, P_{\hat{\mathbf{Y}}}) = \mathbb{E}_{\mathbf{y},\mathbf{y}' \sim P_\mathbf{Y}}[k(\mathbf{y}, \mathbf{y}')] + \mathbb{E}_{\hat{\mathbf{y}},\hat{\mathbf{y}}' \sim P_{\hat{\mathbf{Y}}}}[k(\hat{\mathbf{y}}, \hat{\mathbf{y}}')] - 2 \cdot \mathbb{E}_{\mathbf{y} \sim P_\mathbf{Y}} \mathbb{E}_{\hat{\mathbf{y}} \sim P_{\hat{\mathbf{Y}}}}[k(\mathbf{y}, \hat{\mathbf{y}})], \tag{33}$$

with marginals $P_Y$ and $P_{\hat{Y}}$, representing the true and predictive segmentation distribution, and a distance represented by kernel $k : \mathcal{Y} \times \mathcal{Y} \to \mathbb{R}$, usually the $(1-\text{IoU})$ or $(1-\text{Dice})$ score (squared by the GED). However, solely relying on the GED has been criticized due to undesirable inductive biases (Kohl et al., 2019a; Zepf et al., 2024). Hence, Kohl et al. (2019a) introduce an alternative metric known as Hungarian Matching (HM), which compares the predictions against the ground-truth labels through a cost matrix. This can be also formally denoted as finding the permutation matrix $P$, subject to the objective

$$\text{HM}(\mathcal{Y}, \hat{\mathcal{Y}}) = \frac{1}{N^2} \min_\mathbf{P} \text{Tr}(\mathbf{PM}), \tag{34}$$

where the elements of $M$ are $M_{i,j} = k(\mathcal{Y}_i, \hat{\mathcal{Y}}_j)$, and $N^2$ represents the number of elements within the matrix. Subsequently, the unique optimal coupling between the two sets that minimize the average cost is determined through a combinatorial optimization algorithm. Hu et al. (2022) use the Normalized Cross Correlation (NCC) to measure the similarity between the true and predictive set, which can formally be defined as

$$\text{NCC} = \frac{1}{n\sigma\hat{\sigma}} \sum_i (a_i - \mu) \cdot (\hat{b}_i - \hat{\mu}), \tag{35}$$

with $n$ being the number of pixels, and $\mu, \sigma$ and $\hat{\mu}, \hat{\sigma}$ the mean and standard deviation of the true and predicted segmentations set, respectively. Variables $a$ and $b$ represent the uncertainty map calculated through the pixel-wise variance across the true and predicted segmentations, respectively.

**Approaches** The most straightforward approach is to directly model the empirical stochasticity in the annotations with stochasticity in the features. Given that annotators have an intrinsically associated expertise, it can also be possible to model the annotator distribution in the parameters. However, relying on the model to infer ambiguity in the parameters by observing the data, can become quite burdensome. This observation is reflected in literature, where we see that most approaches use conditional generative models to encapsulate observer variability (See Table 1). While conditional generative models have been successfully applied for the task at hand, it has been shown that such models, without explicit conditioning, do not encapsulate more subtle variations such as distinct labeling styles of annotators (Zepf et al., 2023a). VAE-based Probabilistic U-Net is the most popular approached followed by its hierarchical variants (HPU-Net and PHI-Seg) and SSNs. More recently, the growing popularity of DDPMs is apparent in the field.

Table 1: Methods used for the application of encapsulating Observer Variability.

| Method | Literature |
|---|---|
| PixelCNN | Zhang et al. (2022c) |
| SSN | Monteiro et al. (2020); Kahl et al. (2024); Zepf et al. (2024; 2023a); Philps et al. (2024) |
| GAN | Kassapis et al. (2021) |
| VAE | Gao et al. (2022); Kohl et al. (2019b); Valiuddin et al. (2021); Selvan et al. (2020); Bhat et al. (2023; 2022a); Valiuddin et al. (2024b); Qiu & Lui (2020); Valiuddin et al. (2024a); Viviers et al. (2023c); Schmidt et al. (2023); Long et al. (2021b); Zepf et al. (2023a; 2024); Hu et al. (2022); Viviers et al. (2023b); Savadikar et al. (2021); Philps et al. (2024) |
| HVAE | Zhang et al. (2022b); Rafael-Palou et al. (2021); Kohl et al. (2019a); Baumgartner et al. (2019); Gantenbein et al. (2020) |
| DDPM | Chen et al. (2023a); Zbinden et al. (2023); Rahman et al. (2023) |
| MC dropout | Kahl et al. (2024); Philps et al. (2024) |
| Ensembling | Zepf et al. (2024); Hu et al. (2023) |
| TTA | Kahl et al. (2024) |

## 6.2 Active Learning

The field of Active Learning aims to reduce the costly annotation procedure by careful selection of the most informative samples for training (Settles, 2009; Ren et al., 2021). Instead of randomly labeling data, the model directs expert time toward the most challenging examples, leading to more efficient learning. Active Learning is an extensively researched topic in both traditional (Cohn et al., 1996) and deep Machine Learning (Gal et al., 2017b), including uncertainty-based approaches applied to image-segmentation pairs. In such methods, it is assumed that predictions with high uncertainty provide the most informative samples for training. Moreover, this core idea extends to active perception, where an autonomous agent, like a robot or a drone, physically acts to resolve its own environmental uncertainty. For instance, instead of guessing what an obscured object is, it might move its camera to get a better view. In both cases, the system actively seeks the most valuable information, either from a human annotator or by interacting with the world, to improve its understanding and performance.

**Evaluation**  Selecting the most informative samples for training contains two elements. Firstly, the uncertainty needs to be quantified. This is usually the predictive entropy (Kasarla et al., 2019; Shen et al., 2021; Xie et al., 2022) or variance (Ozdemir et al., 2021), but in some instances the uncertainty is estimated through Bayesian Active Learning by Disagreement (BALD) (Houlsby et al., 2011; Ma et al., 2024b; Shen et al., 2021). Simply stated, the reduction in posterior entropy indicates the informativeness of a data sample. In turn, this can also be formulated by the reduction in predictive entropy in the dual formulation. We can formally denote this as

$$
\begin{aligned}
\mathrm{BALD} \equiv H[\,\boldsymbol{\theta}|\mathcal{D}\,] &- \mathbb{E}_{p(\mathbf{Y}|\mathbf{x}^*,\mathcal{D})}[\,H[\,\boldsymbol{\theta}|\mathbf{y},\mathbf{x}^*,\mathcal{D}\,]\,], \\
&= H[\mathbf{y}|\mathbf{x}^*,\mathcal{D}] - H[\mathbf{y}|\boldsymbol{\theta},\mathbf{x}^*,\mathcal{D}], \\
&= H[\mathbf{y}|\mathbf{x}^*,\mathcal{D}] - \mathbb{E}_{q(\boldsymbol{\theta}|\mathcal{D})}[\,H[\mathbf{y}|\mathbf{x}^*,\boldsymbol{\theta}]\,] = I(\mathbf{y},\boldsymbol{\theta}|\mathbf{x}^*,\mathcal{D}).
\end{aligned}
\tag{36}
$$

Note the similarity between the mutual information $I$ obtained from the predictive entropy decomposition in Equation (3). Secondly, to evaluate whether the active learning strategy is beneficial, the performance is mostly measured subject to increasingly stringent budget requirements (i.e. a percentage of the original dataset) (Yang et al., 2017; Sourati et al., 2018; Kasarla et al., 2019; Mahapatra et al., 2018; Sinha et al., 2019; Siddiqui et al., 2020; Li & Alstrøm, 2020; Kim et al., 2021; Shen et al., 2021; Burmeister et al., 2022; Ma et al., 2024b). Kahl et al. (2024) argue that Active Learning should be evaluated on image-level as humans do not annotate on pixel-level. Assuming a saturated performance on in-distribution data (training

Table 2: Methods used for the application of Active Learning.

| Method | Literature |
|---|---|
| SoftMax | Kasarla et al. (2019); García Rodríguez et al. (2020); Xie et al. (2022); Wu et al. (2021); Burmeister et al. (2022); Gaillochet et al. (2023) |
| SSN | Kahl et al. (2024) |
| Adversarial VAE / GAN | Mahapatra et al. (2018); Kim et al. (2021); Sinha et al. (2019) |
| Ensembling | Yang et al. (2017); Nath et al. (2020); Khalili et al. (2024) |
| MC dropout | Kahl et al. (2024); Gorriz et al. (2017); Ozdemir et al. (2021); Hiasa et al. (2019); Li & Alstrøm (2020); Shen et al. (2021); Ma et al. (2024b); Gaillochet et al. (2023); Siddiqui et al. (2020); Sadafi et al. (2019) |
| TTA | Gaillochet et al. (2023) |

cycle $t_1$), the author propose to measure the improved relative performance, $C$, subject to a second training sample ($t_2$) from selected samples

$$C_{\text{total}} = C_{\text{method}} - C_{\text{random}}, \tag{37}$$

with

$$C = \frac{P_{t_2} - P_{t_1}}{P_{t_1}}, \tag{38}$$

where $C_{\text{method}}$ denotes the relative improvement achieved by the employed method, and $C_{\text{random}}$ is a correction term accounting for gains due to random sampling, and $P$ suggested to be the Dice-score, but can theoretically be any segmentation metric.

**Approaches** As can be seen in Table 2, most of the literature use MC dropout or ensembles. Furthermore, using the plain SoftMax activation has also been commonly used. Often, solely relying on the most uncertain samples can result in samples with limited diversity and therefore samples selected on the *representativeness* (Ozdemir et al., 2021; Shen et al., 2021; Wu et al., 2021; Mahapatra et al., 2018; Sinha et al., 2019). Another crucial modeling element pertains the aggregation of the pixel-level uncertainties. Therefore, closeness to boundary (Gorriz et al., 2017; Kasarla et al., 2019; Ma et al., 2024b) or specific regions (Gaillochet et al., 2023; Wu et al., 2021; Kasarla et al., 2019) are additionally incorporated when combining the uncertainty values across pixels. Siddiqui et al. (2020) determine the entropy through estimating the entropy across viewpoints in multi-view datasets. Gaillochet et al. (2023) argue the superiority of selecting samples based on batch-level uncertainty.

### 6.3 Model introspection

The provided uncertainty can be used to gauge prediction quality of a model. For example, an agricultural drone's segmentation model provides a clear example. When distinguishing "crop" vs. "weed", it will segment healthy fields with low uncertainty, signaling a reliable output. However, for an anomalous patch of stressed plants, it will produce a segmentation with high uncertainty. This high score directly flags the prediction as low-quality and likely erroneous, preventing a costly automated action like misapplying herbicide. As evident, the described problem can also be considered as Out-of-Distribution (OOD) detection (Lambert et al., 2022; Holder & Shafique, 2021).

**Evaluation** The relationship between uncertainty and model accuracy has been formalized by Mukhoti & Gal (2018) through the Patch Accuracy vs Patch Uncertainty (PAvPU) metric. Firstly, the accuracy given a certain prediction, $p(A|C)$, and secondly, the uncertainty given an inaccurate prediction $p(U|I)$. Given a threshold $u_T$ that distinguishes certain from uncertain pixels or patches, we can define pixels that are accurate and certain, accurate and uncertain, inaccurate and certain, inaccurate and uncertain,

Table 3: Methods used for the application of Model Introspection.

| Method | Literature |
|--------|-----------|
| VI | Ng et al. (2022); LaBonte et al. (2019) |
| LA | Zepf et al. (2023b) |
| TTA | Kahl et al. (2024); Wang et al. (2019b); Whitbread & Jenkinson (2022); Roshanzamir et al. (2023); Dahal et al. (2020) |
| Ensembling | Jungo et al. (2020); Mehrtash et al. (2020); Czolbe et al. (2021); Holder & Shafique (2021); Jungo & Reyes (2019); Ng et al. (2022); Hann et al. (2021); Jungo & Reyes (2019); Linmans et al. (2020); Pavlitskaya et al. (2020); Valada et al. (2017) |
| MC Dropout | Jungo et al. (2020); Kahl et al. (2024); Whitbread & Jenkinson (2022); Kendall et al. (2016); Kampffmeyer et al. (2016); Dechesne et al. (2021); Morrison et al. (2019); Qi et al. (2023); Eaton-Rosen et al. (2018); Jungo et al. (2018a;b); Roy et al. (2018a;b); Mehrtash et al. (2020); Nair et al. (2020); Sander et al. (2019); Hasan & Linte (2022b); Camarasa et al. (2021); Roshanzamir et al. (2023); Sedai et al. (2018); Seeböck et al. (2019); DeVries & Taylor (2018); Czolbe et al. (2021); Hoebel et al. (2019); Bhat et al. (2021); Dahal et al. (2020); Antico et al. (2020); Lambert et al. (2022); Hasan & Linte (2022a); Jungo & Reyes (2019); Ng et al. (2022); Iwamoto et al. (2021); Hoebel et al. (2020) |
| SSN | Kahl et al. (2024); Ng et al. (2022) |
| VAE | Chotzoglou & Kainz (2019); Viviers et al. (2023a); Czolbe et al. (2021); Bian et al. (2020) |

denoted by $u_{ac}$, $u_{au}$, $u_{ic}$, $u_{iu}$, respectively. Consequently, the authors combine $p(A|C) = n_{ac}/(n_{ac} + n_{au})$ and $p(U|I) = n_{ic}/(n_{ic} + n_{iu})$ to obtain the Patch Accuracy vs Patch Uncertainty (PAvPU) metric, defined as

$$\text{PAvPU} = \frac{n_{ac} + n_{au}}{n_{ac} + n_{au} + n_{ic} + n_{iu}}. \tag{39}$$

It can be noted that uncertainty is usually only obtained on pixel basis, while crucial information can be present in structural statistics. Hence, Roy et al. (2018b) propose to use the Coefficient of Variation (CoV), which addresses this by measuring structural uncertainty through dividing the volume variance over the mean for all samples. The authors also evaluate structural uncertainties by assuming predictions with the pair-wise average overlap between all respective samples.

$$\overline{\text{D}} = \mathbb{E}_{p_{Y|X^*}} \left[ \text{Dice}(t(y_i), t(y_j)) \right] \quad \forall i \neq j. \tag{40}$$

**Approaches** Considering Table 3, it can be observed that parameter modeling methods such as MC Dropout and Ensembles are by far the most popular choice of uncertainty encapsulation method. However, Concrete dropout (Rumberger et al., 2020b; Mukhoti & Gal, 2018), M-heads (auxiliary networks) (Jungo & Reyes, 2019; Linmans et al., 2020), MoE (Pavlitskaya et al., 2020) Laplace Approximation (Zepf et al., 2023b), Variational Inference in 3D (LaBonte et al., 2019), and test-time augmentation (Dahal et al., 2020) have also been used for this application. Moreover, including information on localized uncertainty to the training objective has shown to improve generalization capabilities (Ozdemir et al., 2017; Bian et al., 2020; Iwamoto et al., 2021; Li et al., 2021).

### 6.4 Model generalization

As mentioned earlier in Section 5.1, sampling new parameter permutations often improves segmentation performance, due to the model combining effect. In contrast to other downstream tasks, evaluation does not require specialized metrics besides quantifying the relative change in model performance. For instance, dropout layers at the deeper decomposition levels of the SegNet (Badrinarayanan et al., 2015) has shown to improve model performance (Kendall et al., 2016). MC Dropout, and specifically Concrete Dropout (Mukhoti & Gal, 2018; Rumberger et al., 2020b)), has also led to improved performance. The improved generalization

Table 4: Methods used for the application of improved Model Generalization.

| Method | Literature |
|---|---|
| SoftMax | Kasarla et al. (2019); García Rodríguez et al. (2020); Xie et al. (2022); Wu et al. (2021); Burmeister et al. (2022); Gaillochet et al. (2023) |
| SSN | Zepf et al. (2023a); Ng et al. (2022) |
| VAE | Viviers et al. (2023a); Zepf et al. (2023a); Hu et al. (2022) |
| DDPM | Wolleb et al. (2021); Wu et al. (2023b;a); Amit et al. (2021); Zbinden et al. (2023); Bogensperger et al. (2023); Hoogeboom et al. (2021) |
| MC dropout | Mukhoti & Gal (2018); Rumberger et al. (2020b); Saha et al. (2021a); Zhang et al. (2022a); Whitbread & Jenkinson (2022); Huang et al. (2018); Mukhoti et al. (2020); Ng et al. (2022); Wickstrøm et al. (2020) |
| Ensembling | Ng et al. (2022); Larrazabal et al. (2021b); Kamnitsas et al. (2018); Saha et al. (2021b); Hu et al. (2023); Ji et al. (2021) |
| TTA | Wang et al. (2019a); Rakic et al. (2024); Whitbread & Jenkinson (2022) |

from ensembling has also shown to produce more calibrated outputs (Mehrtash et al., 2020). Ng et al. (2022) found that ensembling results in the best performance improvement, while BBB is more robust to noise distortions. In other work, orthogonality within and across convolutional filters of the ensemble is enforced through minimizing their cosine similarity, which reaped similar merits (Larrazabal et al., 2021b). Ensembling has also been performed using M-Heads (Hu et al., 2023; Jungo & Reyes, 2019). Nonetheless, individual models in conventional ensembles receive data in an unstructured manner. Hence, specific subsets of the data can also be assigned to particular models with MoEs (Pavlitskaya et al., 2020; Ji et al., 2021; Gao et al., 2022). Generative models, though typically used for other applications, have also led to performance improvements across a wide range of studies (Saha et al., 2021a; Wolleb et al., 2021; Wu et al., 2023b; Amit et al., 2021; Bogensperger et al., 2023; Zbinden et al., 2023; Viviers et al., 2023a; Zepf et al., 2023a; Hu et al., 2022).

> **Key takeaways**: Parameter distribution methods are commonly utilized for Active Learning, Model Introspection, and Model Generalization, while feature distribution modeling is employed for Observer Variability. This observation suggests the irreducibility of uncertainty tied to parameter-level modeling, while reducibility aligns with feature-level modeling.

## 6.5 Application domains

In Table 5, we link the discussed methodologies and tasks to their respective application domains. It is evident that the medical domain, followed by the automotive field, is the most prominent for probabilistic segmentation tasks. The high-stakes nature of both domains contributes to their prominence, as uncertainty-aware models are especially vital in safety-critical applications. Several benchmark datasets are central to these applications. For instance, the LIDC-IDRI dataset (Armato, 2011), with its multiple expert delineations of lung nodules in CT scans, is a key benchmark for quantifying observer variability. In the autonomous driving domain, outdoor scene understanding applications widely utilize the Cityscapes dataset. Similarly, MRI-based research is often advanced by datasets like BraTS or challenges such as QUBIQ (Menze et al., 2021).

Table 5: Overview that links literature to various domains and applications.

| | Observer Variability | Active Learning | Model Introspection | Model Generalization |
|---|---|---|---|---|
| Outdoor scenes | Kahl et al. (2024); Kohl et al. (2019b); Kassapis et al. (2021); Gao et al. (2022) | Sinha et al. (2019); Kahl et al. (2024); Kasarla et al. (2019); Kim et al. (2021); Xie et al. (2022); Wu et al. (2021) | Pavlitskaya et al. (2020); Hasan & Linte (2022b); Sander et al. (2019); Ng et al. (2022); Hasan & Linte (2022a); Mehrtash et al. (2020); Kendall et al. (2016); Kahl et al. (2024); Holder & Shafique (2021); Qi et al. (2023) | Ng et al. (2022); Huang et al. (2018); Zbinden et al. (2023); Mukhoti & Gal (2018); Amit et al. (2021); Hoogeboom et al. (2021); Valada et al. (2017); Rakic et al. (2024) |
| Indoor scenes | | Siddiqui et al. (2020); Wu et al. (2021) | Qi et al. (2023) | |
| Various objects | Philps et al. (2024) | | Kendall et al. (2016); Qi et al. (2023); Morrison et al. (2019) | |
| Remote sensing | | García Rodríguez et al. (2020) | Dechesne et al. (2021); Kampffmeyer et al. (2016); Dechesne et al. (2021) | |
| Microscopy | Kohl et al. (2019a); Zepf et al. (2023a); Schmidt et al. (2023); Philps et al. (2024) | Khalili et al. (2024); Li & Alstrøm (2020); Yang et al. (2017); Sadafi et al. (2019) | Linmans et al. (2020); Iwamoto et al. (2021) | Bogensperger et al. (2023); Amit et al. (2021) |
| Brain MRI | Philps et al. (2024); Bhat et al. (2023); Hu et al. (2023); Philps et al. (2024); Savadikar et al. (2021); Zhang et al. (2022c) | Zepf et al. (2023b); Ma et al. (2024b); Shen et al. (2021) | Roy et al. (2018b); Mehrtash et al. (2020); Jungo & Reyes (2019); Eaton-Rosen et al. (2018); Wang et al. (2019b); Jungo et al. (2020); Whitbread & Jenkinson (2022); Monteiro et al. (2020); Lambert et al. (2022); Jungo et al. (2018b;a) | Wolleb et al. (2021); Wu et al. (2023b;a); Chen et al. (2023a); Wang et al. (2019a); Kamnitsas et al. (2018); Larrazabal et al. (2021b); Wang et al. (2019b); Whitbread & Jenkinson (2022) |
| Prostate MRI | Baumgartner et al. (2019); Hu et al. (2022; 2023); Zepf et al. (2024) | Burmeister et al. (2022); Gaillochet et al. (2023) | Mehrtash et al. (2020); Zepf et al. (2023b) | Hu et al. (2022); Saha et al. (2021a); Rakic et al. (2024); Saha et al. (2021b) |
| Various MRI | Rahman et al. (2023) | Ozdemir et al. (2021); Hiasa et al. (2019); Ng et al. (2022); Burmeister et al. (2022); Ma et al. (2024b); Nath et al. (2020); Gaillochet et al. (2023) | Bhat et al. (2021); Camarasa et al. (2021); Nair et al. (2020); Roshanzamir et al. (2023); Hann et al. (2021) | Ji et al. (2021); Ng et al. (2022) |
| OCT imaging | Selvan et al. (2020) | | Seeböck et al. (2019); Bian et al. (2020); Sedai et al. (2018) | Rakic et al. (2024); Ji et al. (2021); Wu et al. (2023a;b) |
| Dermoscopy | | Li & Alstrøm (2020); Gorriz et al. (2017) | Zepf et al. (2023b); Jungo & Reyes (2019); DeVries & Taylor (2018); Czolbe et al. (2021) | Wu et al. (2023a); Zepf et al. (2023a) |
| Lung CT | Rafael-Palou et al. (2021); Gao et al. (2022); Valiuddin et al. (2024b); Chen et al. (2023a); Zhang et al. (2022b); Monteiro et al. (2020); Qiu & Lui (2020); Kohl et al. (2019a); Hu et al. (2022); Zhang et al. (2022c); Gantenbein et al. (2020); Viviers et al. (2023b;c); Valiuddin et al. (2024a); Zbinden et al. (2023); Zepf et al. (2024); Kahl et al. (2024); Bhat et al. (2022a); Long et al. (2021b); Valiuddin et al. (2021); Rahman et al. (2023); Kohl et al. (2019b); Hu et al. (2023); Bhat et al. (2023); Kassapis et al. (2021); Long et al. (2021a); Selvan et al. (2020); Baumgartner et al. (2019) | Kahl et al. (2024); Hoebel et al. (2020) | Kahl et al. (2024); Czolbe et al. (2021); Chotzoglou & Kainz (2019) | Hu et al. (2022); Rakic et al. (2024); Zhang et al. (2022a) |
| Various CT | Valiuddin et al. (2024b); Hu et al. (2023) | Hiasa et al. (2019) | Hoebel et al. (2019); LaBonte et al. (2019) | Viviers et al. (2023a) |
| Various US | Rahman et al. (2023) | Yang et al. (2017) | Antico et al. (2020); Dahal et al. (2020); Bian et al. (2020) | Wu et al. (2023b;a); Rumberger et al. (2020b) |
| Others | Valiuddin et al. (2021) | Mahapatra et al. (2018) | | Wickstrøm et al. (2020) |

# 7 Discussion

This section builds on the reviewed theory and literature to guide researchers in selecting methods suited to their goals. Furthermore, it highlights key gaps and limitations, offering insights to address current challenges and identify promising directions for future research.

---

**Main discussion points**:
- The interpretation of epistemic and aleatoric uncertainty remain complex and warrant careful reconsideration, despite the literature often suggesting a clear distinction (Section 7.1).

- Challenges related to spatial coherence remain unaddressed (Section 7.2).

- The application-driven field lacks standardization across the downstream tasks, hindering benchmarking and further progress (Section 7.3).

- Uncertainty-based Active Learning is promising and has many open challenges (Section 7.4).

- There exist various limitations and trade-offs across methods (Section 7.5).

- Understanding the critical influence of data characteristics on uncertainty models remains a relatively under-explored area of research (Section 7.6).

- The field of segmentation could advance significantly by treating the backbone as a critical modeling choice and embracing contemporary architectures (Section 7.7).

- The segmentation community should actively explore and integrate novel uncertainty quantification methodologies (Section 7.8).

- Uncertainty modeling should address the growing importance of multiclass, instance, and panoptic segmentation problems (Section 7.9).

---

## 7.1 Disentangling uncertainties

Proper disentanglement of the uncertainties allows to accredit high-entropy outputs to either model ignorance about the data (epistemic), or inherent noise in the data generation process (aleatoric). As Equation (2) implies, this can be achieved by introducing stochasticity into either the model's parameters or its features. This fundamental distinction not only provides the organizing principle for our subsequent review but also aligns with the approaches empirically observed in the literature. Seemingly straightforward, sufficient literature and discussion in the scientific community exist related to the applicability of this taxonomy in practical cases. As paraphrased from Kirchhof et al. (2025), "*definitions of uncertainty resemble the shapes of clouds — clearly defined from a distance, but losing clarity during approach by dissolving in one another*". We further explore this, highlighting key nuances in uncertainty modeling and contextualize this to segmentation problems. To begin, we first establish the original definitions of these concepts to better understand why these concepts have lead to incorrect definitions, claims and general confusion.

**Aleatoric uncertainty**    reconsiders the non-deterministic relationship between $x \in \mathcal{X}$ and $y \in \mathcal{Y}$, which implies that

$$p(\mathbf{Y}|\mathbf{X}) = \frac{p(\mathbf{Y}, \mathbf{X})}{p(\mathbf{X})} \neq \delta(\mathbf{Y} - F(\mathbf{X})), \tag{41}$$

with Dirac-delta function $\delta$, and mapping $F : \mathcal{X} \rightarrow \mathcal{Y}$. This relationship is characterized by the ambiguity in $\mathbf{X}$ and is inherently probabilistic, due to various reasons such as noise in the data (occlusions, sensor noise, insufficient resolution, etc.) or variability within a class (e.g. not all cats have tails). Hence, the observance of substantial aleatoric uncertainty can in some cases be inevitable, but may also signal the need for higher-quality data acquisition or shifting to another modality. The possible input dependency of the uncertainty develops into further categorization of either *heteroscedastic* (input dependent) or *homoscedastic* (input independent) aleatoric uncertainty.

**Epistemic uncertainty** arises from model ignorance (i.e. related to the parameters). Consequently, epistemic uncertainty should not only be quantified but ideally also minimized. Epistemic uncertainty can be further categorized into two distinct types (Hüllermeier & Waegeman, 2021). The first type pertains uncertainty related to the capacity of the model. For example, under-parameterized models or approximate model posteriors, that can become too stringent to appropriately resemble the true posterior. The ambiguity on the best parameters given the limited capacity induces uncertainty in the learning process, which is also known as *model uncertainty*. Nevertheless, given the complexity of contemporary parameter-intensive CNNs, the model uncertainty is often assumed to be negligible. A more significant contribution to the epistemic uncertainty is due to the limited data availability, known as *approximation uncertainty*, and can often be reduced by collecting more data. Additionally, we have found it to be common to interchangeably use the total epistemic and solely model uncertainty, especially in the context of Monte Carlo Dropout methods (Kendall et al., 2016; Roy et al., 2018b; DeVries & Taylor, 2018; Mehrtash et al., 2020; Burmeister et al., 2022; Iwamoto et al., 2021). To prevent confusion in already established terminology, the term *model uncertainty* should be exclusively used for the induced ambiguity due to limited model complexity, rather than to denote general or the entire epistemic uncertainty.

**Task dependency** It is essential to consider that interpretation of the uncertainty source often differs depending on the modeling context. For example, epistemic uncertainty to be considered as the number of required models to fit the data (Wimmer et al., 2023), evident in ensembling and M-head approaches for tasks related to model introspection and generalization (Sections 6.3 and 6.4). Another perspective pertains model disagreement (Houlsby et al., 2011; Gal et al., 2017b; Kirsch, 2024), relating to the source of observer variability (Section 6.1) and methods that employ mixture-of-expert approaches (Section 5.1.2). Also, it has been simply considered to be the remainder after subtracting the aleatoric from the total uncertainty (Depeweg et al., 2018), such as commonly done in Active Learning (Section 6.2). Aleatoric uncertainty also knows several definitions such as being the Bayes-optimal residual risk (Apostolakis, 1990; Helton, 1997), but remains consistent in related literature as the inherent, irreducible noise in the data inducing variance in the ground-truth distribution (Section 6.1). To avoid confusion, this task-dependency should be considered when interpreting uncertainty.

**Model dependency** Moreover, past literature as well as our overview indicate the influence of specific design choices on the nature of the modeled uncertainty (Der Kiureghian & Ditlevsen, 2009). For instance, the expected entropy w.r.t. a stochastic parameter variable is often assumed to represent the epistemic uncertainty as shown in Equation (3). However, when employing generative models, this behavior changes. As argued by Kahl et al. (2024), the predictive uncertainty can be decomposed as

$$H[\mathbf{Y}|\mathbf{x},\boldsymbol{\theta}] = \underbrace{I[\mathbf{Y},\mathbf{Z}|\mathbf{x},\boldsymbol{\theta}]}_{\text{aleatoric}} + \overbrace{\mathbb{E}_{p(\mathbf{z})}\big[\,H[\mathbf{Y}|\mathbf{Z},\mathbf{x},\boldsymbol{\theta}]\,\big]}^{\text{epistemic}}. \tag{42}$$

In contrast to Equation (3), the mutual information term here encapsulates the aleatoric uncertainty, while the expected entropy reflects the epistemic uncertainty. This is because generative models operate under the assumption that the latent variable $\mathbf{Z}$ captures the inherent data ambiguity, thus representing the aleatoric uncertainty. The mutual information describes how much about this aleatoric uncertainty can be learned by observing the output $\mathbf{Y}$. If one were to know the optimal latent code $\mathbf{Z}$, the mutual information would be zero. In this case, all stochasticity in the output $\mathbf{Y}$ is perfectly represented by $\mathbf{Z}$, meaning the aleatoric uncertainty has been fully captured. Under this view, if all data uncertainty can be fully encapsulated by the latent variable $\mathbf{Z}$, there is no need to model this irreducible stochasticity in the predictive model $p(\mathbf{Y}|\mathbf{Z},\mathbf{x},\boldsymbol{\theta})$, implying it corresponds to the epistemic uncertainty. This is consistent with the principle that epistemic uncertainty is the residual component isolated by subtracting the aleatoric uncertainty from the total predictive entropy. Hence, the calibration network of the GAN-based CAR model of Kassapis et al. (2021) (discussed in Section 4.2.1) is actually more inclined to model the epistemic rather than aleatoric uncertainty. This dependency is also demonstrated by the conflicting ideas on uncertainty encapsulated with TTA. In some works, it is argued that this approach models the aleatoric uncertainty in a better way (Wang et al., 2019a; Ayhan & Berens, 2022; Zhang et al., 2022a; Wang et al., 2019b; Rakic et al., 2024;

Whitbread & Jenkinson, 2022; Roshanzamir et al., 2023). However, literature opposing this also exist and even suggests that TTA enables the modeling of epistemic uncertainty (Hu et al., 2022). Kahl et al. (2024) draw parallels with BNNs and demonstrate that

$$H[\mathbf{Y}|\mathbf{x},\boldsymbol{\theta}] = \underbrace{I[\mathbf{Y},\mathbf{t}|\mathbf{x},\boldsymbol{\theta}]}_{\text{epistemic}} + \overbrace{\mathbb{E}_{\mathbf{t}}[\,H[\mathbf{Y}|\mathbf{t},\mathbf{x},\boldsymbol{\theta}]\,]}^{\text{aleatoric}}. \tag{43}$$

where $t$ are sampled augmentations from the input space $\mathcal{T}$. A well-trained model should be invariant to data augmentations. Consequently, predictive variance across augmentations is treated as irreducible aleatoric uncertainty, which is quantified by a non-zero expected entropy over these predictions. Non-zero mutual information indicates the model has not yet learned invariance to the augmentations and can be reduced with further training, deeming it to be epistemic. This decomposition is analogous to the one for conventional BNNs (Equation 3), while it contrasts with the decomposition used for generative models (Equation 42). The key message thus far is that, after considering the type of uncertainty to model, it is equally important to carefully choose the appropriate method for its representation, as confusion and misunderstandings can cause deviation from the targeted uncertainty.

**Intertwinedness**   Despite the seemingly rigorous and well-defined distinction, it can be seen that implementation rapidly blurs this notion. A systematic study has found that the ability to separate the uncertainties can strongly depend on the data (Kahl et al., 2024). These findings are also supported by research presenting a strong correlation between epistemic and aleatoric uncertainty quantification methods (de Jong et al., 2024; Mucsányi et al., 2024), indicating that they represent uncertainty of similar kind. Kendall & Gal (2017) have also observed that explicitly modeling a single uncertainty tends to compensate for the lack of the other. In probabilistic models, we can also find that the parameter modeling methods, traditionally used for epistemic uncertainty, have been used to model observer variability, by for example using MC-Dropout or ensembling (see Table 1). Furthermore, using a combination of distribution and paramter modeling techniques can also beneficial (Gao et al., 2022). These findings further highlight the notion of the uncertainties being intertwined. On a slight side note, it is notable that popular BNN methods such as the Local Reparameterization trick use stochasticity in the output features of intermediate layers (usually considered to encapsulate aleatoric uncertainty) as a proxy to model the weight distributions (Kingma et al., 2015), further supporting this argument.

**Modeling vs. quantifying**   A common misconception pertains that modeling a specific uncertainty also implies that the predicted output distribution represents that same uncertainty. This undermines the entropy decomposition in Equation (3) distinguishing between encapsulating, i.e. targeting an intended uncertainty type, versus quantifying, i.e. measuring the modeled uncertainty in isolation. For example, the estimation of mutual information is required to express the epistemic uncertainty in isolation. Nonetheless it is common for papers to quantify the uncertainty by inspecting the output distribution subject through repetitive sampling from the parameters densities, thereby approximating the predictive entropy encapsulating both uncertainties (Camarasa et al., 2021; Seeböck et al., 2019; Roshanzamir et al., 2023; Roy et al., 2018a; Rumberger et al., 2020b; Hoebel et al., 2022; Seeböck et al., 2019; Wundram et al., 2024b; Rumberger et al., 2020a; Sedai et al., 2018). This similarly holds for literature emphasizing aleatoric uncertainty when estimating the predictive variance (Zhang et al., 2022c; Monteiro et al., 2020; Valiuddin et al., 2024b; Gao et al., 2022). Since predictive variance can encompass both uncertainties, a crucial distinction must be made. Targeting a specific uncertainty for a task through either parameter or feature modeling is different from truly quantifying a type in isolation, which may require separate validation.

## 7.2   Spatial coherence

The inter-pixel dependency of segmentation masks often introduces additional complexity in proper uncertainty quantification, questioning the commonly repurposed techniques from conventional image-level classification. In this section, we further dive into important details and challenges arising due to the added dimension(s) in both 2D and 3D data.

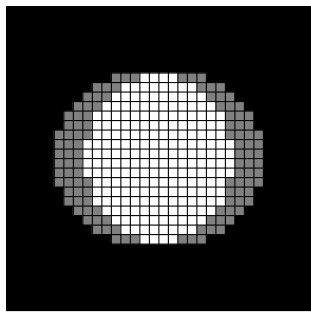 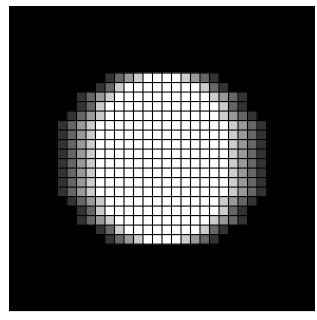 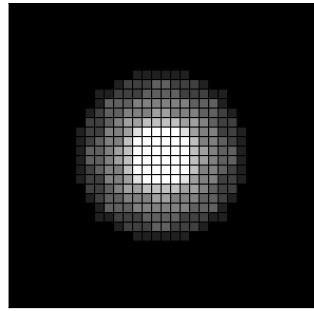

(a) 2 states, true entropy $log_2(2) = 1$ `bit`, factorized entropy $-p_i \sum \log_2 p_i \approx 112$ `bits`

(b) 5 states, true entropy $log_2(2) \approx 2.32$ `bits`, factorized entropy $-p_i \sum \log_2 p_i \approx 114$ `bits`

(c) 8 states, true entropy $log_2(8) = 3$ `bits`, factorized entropy $-p_i \sum \log_2 p_i \approx 190$ `bits`

Figure 13: This figure illustrates why factorized entropy is a poor proxy for the true uncertainty in structured images. The common method to estimate this entropy is by averaging multiple sampled binary segmentation masks, where the resulting pixel intensity indicates the average confidence value. The shapes shown possess only translational uncertainty. The factorized model fails to capture the object's spatial coherence and leads to a significant overestimation of the true entropy.

**Aggregation methods** The mutual information is often utilized to express the reducible, epistemic uncertainty. This requires subtracting the expected entropy from the total entropy of the predictive distribution (see equations (3) and (36)). In such cases, the entropy is usually estimated via mean or summation over all pixel-level uncertainty values (Mukhoti & Gal, 2018; Camarasa et al., 2021; Bhat et al., 2022b; Mukhoti et al., 2020; Ma et al., 2024b; Shen et al., 2021; Li & Alstrøm, 2020; Dechesne et al., 2021; Huang et al., 2018; Nair et al., 2020; Zepf et al., 2023b). However, this asserts the assumption of a factorized categorical distribution, i.e. uncorrelated pixels, which is a strong assumption that ignores the core rationale for the field's existence. Hence, spatial coherence is a fundamental consideration and is frequently neglected in the uncertainty estimation frameworks used for Active Learning.

The problem with factorized entropy can be demonstrated with a simple example: a binary segmentation of a circle on a 32×32 grid. If we assume that the 'latent' uncertainty is purely translational, Figure 13 visualizes the example distributions of possible segmentation maps. The true entropy of the generating source scales with the number of possible states $N$ as $\log_2(N)$. However, using a factorized approach severely overestimates the latent, true uncertainty. This behavior is a direct consequence of the subadditivity property of entropy, which states that

$$H[X_1, X_2, ..., X_D] \leq \sum_i^D H[X_i].  \tag{44}$$

Importantly, this fallacy could explain failure cases of entropy-based uncertainty evaluation methods in tasks such as Active Learning where sample informativeness/diversity is crucial. Elements such as object class dependency, size, shape, proximity, contextual information etc. can all influence the decision-making of the aggregation method. Notably, Kahl et al. (2024) argue that pixel aggregation should be considered as a separate, distinct modeling choice in probabilistic segmentation. The authors demonstrate that naive summation-based aggregation techniques (Wang et al., 2019a; Jungo et al., 2020; Zhang et al., 2022a; Czolbe et al., 2021; Camarasa et al., 2021) can cause the foreground object size to correlate with the uncertainty score. Some empirical studies have explored other types of aggregation (Camarasa et al., 2021; Roy et al., 2018b; Jungo et al., 2020; Kasarla et al., 2019; Xie et al., 2022; Wu et al., 2021). However, we advocate for deeper investigation into well-informed image-level aggregation strategies and their implications for uncertainty quantification.

**Volumetric uncertainty modeling** Processing volumetric clinical data knows two dominant strategies. The first, 2D slice processing, analyzes the volume as a stack of individual images, preserving full context within each slice but severing spatial continuity along the third dimension. The second, 3D patch processing, divides the volume into smaller ones, which maintains local 3D information but loses the global anatomical structure. As compute power increases, end-to-end 3D segmentation models are gaining traction because they can analyze increasingly more volume, preserving spatial continuity and reducing the inconsistencies inherent in these slice- or patch-based methods (Liu et al., 2025; Burmeister et al., 2022; Viviers et al., 2023b; Monteiro et al., 2020). As we have seen the implications of the additional dimension in 2D uncertainty, we similarly argue that 3D segmentation introduces or exacerbates previously mentioned issues related to entropy estimation and aggregation of pixels. Probabilistic segmentation has been employed on 3D data but often are straightforward extensions of patch-wise segmentation architectures (Viviers et al., 2023b; Chotzoglou & Kainz, 2019; Long et al., 2021b; Saha et al., 2021a; 2020; Hoebel et al., 2019). Limited diversity of uncertainty-based sample selection in Active Learning requires more focus, since 2D slices of 3D volumes are often highly redundant (Shen et al., 2021; Burmeister et al., 2022). Furthermore, given the high computational cost of many architectures, we emphasize the need for efficiency with higher-dimensional input tensors. Moreover, methodologies need to be developed in order to appropriately compare 2D to 3D models. For example, a valid volumetric GED cannot be estimated with 2D models. Hence, general guidelines related to translating 2D segmentation models to 3D can be of great benefit for practitioners considering to phase out patch-based approaches for volumetric data.

## 7.3 Standardization

The widespread success of contemporary machine learning models can be largely attributed to open-source initiatives and rigorous benchmarking practices. The field of probabilistic segmentation falls short on this. A key factor contributing to these diverging standards is that the literature is predominantly application-driven, uncertainty often considered merely an auxiliary tool, leading to a fragmented landscape where studies rely on varied datasets and methodologies without a shared evaluation protocol. This problem has been partly addressed by contemporary research attempting to standardize the field through a unified framework after identifying key pitfalls (Kahl et al., 2024). Our manuscript further contributes by introducing coherence and a unifying synthesis across the distinct disciplines of applications, tasks, and methodological developments, with these concepts being further detailed in our recommendations in Section 8.

Despite the aforementioned points, efforts to quantify observer variability show some implicit consensus. Notably, some of the recommendations of Kahl et al. (2024) diverge from these practices, such as how to employ the widely used LIDC-IDRI dataset (Armato, 2011) or the Cityscapes Cordts et al. (2016a) dataset, suggesting a more rigorous examination of this application. Furthermore, the evaluation metrics used across various studies also diverge substantially. While GED and Hungarian Matching are commonly applied to assess model performance, key details, such as the choice of distance kernel, the number of predictive samples, and the handling of correct empty segmentations (which cause NaNs), vary widely. This inconsistency is particularly evident in the work of Zbinden et al. (2023), who attempt to identify common ground across benchmarks but find over 10 (!) different evaluation methods. Additionally, many reported results are not taken from original papers but from follow-up studies using them as baselines. See Table 6 for a comparison of benchmarks on the LIDC-IDRI dataset.

## 7.4 Future of Active Learning

Among the applications considered, uncertainty quantification emerges as a promising approach to Active Learning, showing true potential of potentially reducing annotation costs while preserving or even improving model generalization. Active Learning directly addresses the labeling bottleneck through uncertainty-driven selection, aligning with model introspection and fostering human-AI collaboration in specialized domains. Furthermore, when combined with federated learning, this field can accelerate privacy-centric collaboration by enabling efficient, human-in-the-loop improvement of safety-critical models across decentralized data silos. Similarly, these principles can also translate tp Reinforcement Learning, where uncertainty guides an agent's exploration policy to accelerate learning in complex environments. However, key limitations remain and require further research.

Table 6: Comparison of test evaluations on two versions of the LIDC-IDRI dataset. Table extended from (Zbinden et al., 2023). *Indicates evaluation results not obtained from the original work.

| Method | LIDCv1 | | LIDCv2 | |
|---|---|---|---|---|
| | $GED_{16}$ | $HM\text{-}IoU_{16}$ | $GED_{16}$ | $HM\text{-}IoU_{16}$ |
| PU-Net (Kohl et al., 2019b) | $0.310 \pm 0.010^*$ | $0.552 \pm 0.000^*$ | $0.320 \pm 0.030^*$ | $0.500 \pm 0.030^*$ |
| HPU-Net (Kohl et al., 2019a) | $0.270 \pm 0.010^*$ | $0.530 \pm 0.010^*$ | $0.270 \pm 0.010^*$ | $0.530 \pm 0.010^*$ |
| PhiSeg (Baumgartner et al., 2019) | $0.262 \pm 0.000^*$ | $0.586 \pm 0.000^*$ | - | - |
| SSN (Monteiro et al., 2020) | $0.259 \pm 0.000^*$ | $0.558 \pm 0.000^*$ | - | - |
| CAR (Kassapis et al., 2021) | - | - | $0.264 \pm 0.002$ | $0.592 \pm 0.005$ |
| JProb. U-Net (Zhang et al., 2022d) | - | - | $0.262 \pm 0.000$ | $0.585 \pm 0.000$ |
| PixelSeg (Zhang et al., 2022c) | $0.243 \pm 0.010$ | $0.614 \pm 0.000$ | $0.260 \pm 0.000$ | $0.587 \pm 0.010$ |
| MoSE (Gao et al., 2022) | $0.218 \pm 0.001$ | $0.624 \pm 0.004$ | - | - |
| AB (Chen et al., 2022) | $0.213 \pm 0.001^*$ | $0.614 \pm 0.001^*$ | - | - |
| CIMD (Rahman et al., 2023) | $0.234 \pm 0.005^*$ | $0.587 \pm 0.001^*$ | - | - |
| CCDM (Zbinden et al., 2023) | $0.212 \pm 0.002$ | $0.623 \pm 0.002$ | $0.239 \pm 0.003$ | $0.598 \pm 0.001$ |
| BerDiff (Chen et al., 2023a) | - | - | $0.238 \pm 0.010$ | $0.596 \pm 0.000$ |
| MedSegDiff (Wu et al., 2023b) | - | - | $0.420 \pm 0.030^*$ | $0.413 \pm 0.030^*$ |
| SPU-Net (Valiuddin et al., 2024b) | - | - | $0.327 \pm 0.003$ | $0.560 \pm 0.005$ |

**Model sensitivity and baselines**  Active Learning requires a well-generalized model trained on limited data, rendering performance highly sensitive to budget constraints, model architecture, hyperparameters, and regularization (Mittal et al., 2019; Kirsch, 2024; Munjal et al., 2022; Kirsch et al., 2019; Gaillochet et al., 2023). Furthermore, random sampling, particularly when combined with MC Dropout, often proves to be a surprisingly strong baseline that is difficult to outperform (Kahl et al., 2024; Mittal et al., 2019; Lüth et al., 2023; Burmeister et al., 2022; Siddiqui et al., 2020; Kim et al., 2021; Sinha et al., 2019; Nath et al., 2020), especially with highly imbalanced data (Ma et al., 2024a). This observation can partly be attributed to the fact that MC Dropout likely fails to capture true Bayesian uncertainty (see Section 7.5), inviting exploration of other techniques.

**Sample informativeness**  A set of highly uncertain samples may share similar semantic content, leading the model to overfit a narrow and homogeneous region of the data distribution during early training. Hence, sample selection should be based on both uncertainty and diversity (Ozdemir et al., 2021; Wu et al., 2021; Jensen et al., 2019; Burmeister et al., 2022). Ozdemir et al. (2021) leverage the latent space of a VAE to select representative samples, whereas other studies use feature vectors from prediction models (Wu et al., 2021; Burmeister et al., 2022). Furthermore, the sample diversity offered by such models can be effectively combined with adversarial training (Mahapatra et al., 2018; Sinha et al., 2019).

**Uncertainty calibration and aggregation**  As discussed in Section 4.1, using the predictive entropy of SoftMax-based uncertainty requires proper calibration. Unfortunately, this is rarely verified in practice (Kasarla et al., 2019; García Rodríguez et al., 2020; Xie et al., 2022; Wu et al., 2021; Burmeister et al., 2022; Gaillochet et al., 2023). Furhtermore, most works rely on heuristics to aggregate the pixel-level information such as closeness to edges, boundaries or the use of super-pixels (Gorriz et al., 2017; Kasarla et al., 2019; Ma et al., 2024b; Gaillochet et al., 2023; Wu et al., 2021; Kasarla et al., 2019; Bengar et al., 2021). Moreover, the mutual information is often quantified by plain summation (i.e. assuming pixel independence) (Shen et al., 2021; Ma et al., 2024b). We hypothesize that these ad-hoc aggregation techniques, as mentioned in Section 7.2, are a key limiting factor to uncertainty-based Active Learning.

**The Cold-Start Problem**  Finally, the "cold start" problem (Nath et al., 2022; Houlsby et al., 2014; Yuan et al., 2020a), the notion that uncertainty methods work poor on the initial samples, is rarely addressed. Most approaches assume a well-trained model is already available, yet methods like MC Dropout require warm-up epochs to yield meaningful uncertainty. Addressing this gap, post-hoc techniques such as that of

Sourati et al. (2018), which enable uncertainty estimation from pretrained deterministic networks, represent a valuable and underexplored direction for early-stage sample selection.

## 7.5 Method suboptimality

To provide a necessary contrast to the benefits of many models, this section discusses key methodological limitations. These limitations will serve as guidance for readers selecting the most optimal methods and will be utilized when we present our recommendations in Section 8.

**Invertible covariance of SSNs**  Stochastic Segmentation Networks (SSNs), operate by placing a multivariate gaussian over the classification head of an existing model. However, they can suffer from training instability primarily due to the requirement that the covariance matrix remain invertible. To address this critical limitation, the authors propose two distinct remedial strategies: masking out the background to prevent exploding variances and employing uncorrelated gaussians when the covariance matrix becomes singular, which may impact model performance and theoretical purity.

**GAN training**  The Calibrated Adversarial Refinement network (CAR, as treated in Section 4.2.1), on the other hand, is a less commonly used method in literature. This can perhaps be accredited due to the well-known instability of GAN-based models (Arjovsky & Bottou, 2017). GANs often require additional heuristic terms in the loss function to ensure stable training. CAR is no exception to this, as its objective function combines four separate losses. The multiplicity of individual losses is critical to the model's success. Nonetheless, this complex objective function suggests the training process may pose substantial fine-tuning challenges, potentially explaining the model's limited practical adoption.

**Mode collapse with ELBOs**  Known for its flexibility to various datasets, fast sampling time and the interpretable latent space, Variational Autoencoders (VAEs) seem to be the most popular choice for the task of encapsulating Observer Variability. (See Table 1 in Section 6.1). Nonetheless, the shortcomings of VAEs are well known. For example, such models suffer from inference suboptimality related to ELBO optimization (Cremer, 2018; Zhao et al., 2017) and literature on the VAE-based PU-Net often describe behavior similar to the well-known phenomena of *mode collapse* (Valiuddin et al., 2024a;b; Qiu & Lui, 2020). This has been hypothesized to be caused by excessively strong decoders (Chen et al., 2016) and is especially apparent when dealing with complex hierarchical decoding structures, where additional modifications such as the GECO objective (Kohl et al., 2019a), residual connections (Kohl et al., 2019a; Baumgartner et al., 2019), or deep supervision (Baumgartner et al., 2019) are required for generalization.

**Limitation of sequential modeling**  The adoption of sequential sampling models DDPMs or PixelSeg is rather limited compared to the VAE-based models. This is unexpected, as both models outperforms the latter. This is likely due to their tedious sequential inference procedure (Song et al., 2020; Zheng et al., 2023; Meng et al., 2023; Zhang et al., 2022c), a crucial limitation that is exacerbated in supervised settings, which often require validation through sampling on a separate data split. Despite these shortcomings, the strengths of DDPMs should not be overlooked. Their iterative sampling allows for highly flexible modeling and preserves high-frequency details often lost in latent-variable approaches like VAEs, which can produce blurry reconstructions.

**Discrete vs. continuous**  It can be noted in Table 6 that the best-performing Denoising Diffusion Probabilistic Models (DDPMs) are discrete in nature (Chen et al., 2023a; Zbinden et al., 2023). While it is debatable whether shifting to categorical distributions is required for complex image generation (Chen et al., 2022). This can visually be already quite apparent for the multi-class case, where the transition to noise is visually more gradual in the discrete transition (see Figure 14). However, Table 6 shows AB (Chen et al., 2022) performs almost identically to CCDM (Zbinden et al., 2023), suggesting that straightforward thresholding of continuous models have little-to-no performance sacrifice compared with explicitly modeling discrete distributions. These observations suggests that the modeling intuition of a categorical latent space, despite its fundamental soundness, lacks demonstrable merit.

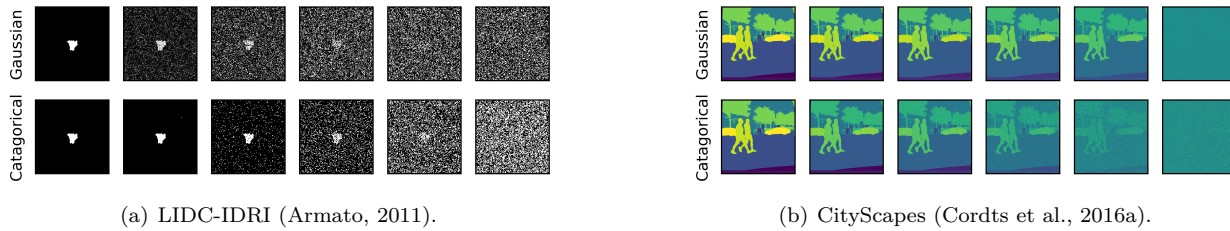

(a) LIDC-IDRI (Armato, 2011).

(b) CityScapes (Cordts et al., 2016a).

Figure 14: Continuous vs. categorical forward diffusion process with cosine noise scheduling (Nichol & Dhariwal, 2021). Note the use of categorical diffusion that results in a more gradual transition for the multi-class case.

**Criticism of MC Dropout** The majority of literature on parameter distribution modeling appears to rely on approximations of variational inference (VI). Among these, MC Dropout is the most commonly used due to its simplicity, low computational cost, and ease of implementation. However, MC Dropout has faced substantial criticism (Folgoc et al., 2021; Osband, 2016; Kingma et al., 2015; Gal et al., 2017a). For instance, it can assign zero probability to the true posterior or introduce erroneous multi-modality (Folgoc et al., 2021). It is also sensitive to model size and dropout rate rather than the observed data (Osband, 2016; Kingma et al., 2015). To address these issues, some alternatives attempt to learn the dropout rate (Kingma et al., 2015; Gal et al., 2017a). In cases of improved Model Generalization (Section 6.4), it is also debatable whether the performance gains stem from better uncertainty modeling or from other factors. For instance, both ensembling and MC Dropout introduce a model combination effect, effectively averaging multiple networks. MC Dropout also resembles placing an $L_2$ penalty on weights (Gal & Ghahramani, 2016). While these effects may improve generalization, similar gains might be achieved with simpler, more efficient regularizers. This suggests that perhaps it is the inherent stochasticity of the methods that is the key beneficial factor. Still, if achieving peak performance is the primary goal, uncertainty modeling is likely not the most efficient route due to its added computational cost. Instead, the improved performance may simply be an unintended side effect of techniques chosen for a different purpose.

## 7.6 Data dependency

In contrast to many other domains within computer vision, this field exhibits a notably greater absence of a single state-of-the-art method. Performance is heavily contingent upon multiple factors such as modeling context and application. However, data dependency often remains unexamined. Especially in cases where parameter distributions are modeled. For instance, MC dropout was found to perform better than ensembling (Hoebel et al., 2020; Roy et al., 2018b), while in other works ensembling excels (Mehrtash et al., 2020; Ng et al., 2022). Also, there is convincing evidence to prefer Concrete Dropout rather than MC Dropout (Mukhoti & Gal, 2018). All things considered, the preference for a particular methodology seems to carry a strong data-dependency (Dahal et al., 2020; Jungo & Reyes, 2019; Burmeister et al., 2022; Ng et al., 2022; Kahl et al., 2024). For feature modeling, some models in Table 6 may appear to be overall best performers. However, their validation is often limited to binary, multi-annotated data, thereby compromising any claims of generalization to other data types, such as multi-class or single-annotated scenarios. Current research often focuses on incorporating uncertainty into already high-performing systems rather than comparing distinct methods. This trend necessitates a more rigorous, data-driven perspective to validate and understand the optimal uncertainty modeling strategy. Hence, a valuable benchmark study should comprehensively compare all available methods (including both feature and parameter modeling approaches) across all relevant task with special focus on the data dependencies.

## 7.7 Segmentation backbone

Due to emphasis on the method, task and application, the backbone feature extractor is often an overlooked element in uncertainty modeling. For example, CNN-based encoder-decoder models such as the U-Net, remain the preferred backbone architecture (Eisenmann et al., 2023). Nevertheless, Vision Transformers

(ViTs) (Dosovitskiy et al., 2020) such as SegFormer, Swin U-Net, Mask2Former already outperform CNNs in general segmentation problems (Xie et al., 2021; Zhang & Yang, 2021; Cheng et al., 2022). With the exception of experiments in Kahl et al. (2024) (combining SSN (Monteiro et al., 2020) with HRNet (Yuan et al., 2020b)), all employed models use a CNN backbone. The limited adoption of ViTs likely stems from CNNs' advantageous inductive biases, whereas Transformers typically require extensive pretraining on large datasets (Caron et al., 2021). Nonetheless, CNNs have benefited from the recent developments in Transformers. For instance, *ConvNext* takes inspiration from contemporary Transformers to modernize existing ResNet-based CNNs, retaining the inductive biases of convolutional filters and achieving significant performance gains (Liu et al., 2022c). These same inductive biases have been added to ViTs through CNN adapters to also enhance performance (Chen et al., 2023b). Building upon these observations, we conclude that future research should actively explore the integration of state-of-the-art Vision Transformer (ViT) and hybrid CNN-Transformer backbones to advance the field.

### 7.8 Contemporary uncertainty

While current efforts frequently leverage established uncertainty modeling techniques for segmentation, the practical integration involves significant challenges related to ensuring architectural and task compatibility. To accelerate progress, the segmentation community should actively explore and integrate newly developed uncertainty quantification methodologies, rather than relying solely on standard, established practices.

**VI alternatives**   Approaches besides VI, such as Markov Chain Monte Carlo (MCMC) or Laplace Approximations, are also viable options to approximate the Bayesian posterior. Especially the Laplace Approximation can be very beneficial, as it is easily applicable to pretrained networks. Notably, both Laplace Approximation and VI are biased and operate in the neighborhood of a single mode, while MCMC methods are a more attractive option when expecting to fit multi-modal parameter distributions. Zepf et al. (2023b) explored the impact of the Laplace Approximation on segmentation networks. However, beyond this work, the method has received limited attention in the field.

**Sampling-free uncertainty**   The multiple forward passes required in many Bayesian uncertainty quantification can incur cumbersome additional costs. Hence, considerable efforts have been made towards sampling-free uncertainty models (Mukhoti et al., 2023; Liu et al., 2020; Van Amersfoort et al., 2020; Postels et al., 2019), only depending on a single forward pass. Mukhoti et al. (2023) show that Gaussian Discriminant Analysis after training with a SoftMax predictive distribution can in some instances surpass methods such as MC Dropout and ensembling. Along similar lines, Evidential Deep Learning also possesses the advantage of quantifying both uncertainties with a single forward pass. This framework is based on a generalization of Bayes theorem, known as the Dempster-Shafer Theory of Evidence (DST) (Dempster, 1967). While common in Bayesian probability, DST does not require prior probabilities and bases subjective probabilities on belief masses assigned on a frame of discernment, i.e. the set of all possible outcomes. The use of Evidential Deep Learning has seen success in conventional classification problems (Sensoy et al., 2018), and Ancha et al. (2024) recently applied this concept to segmentation attempt to decouple aleatoric and epistemic uncertainty within a single model. However, follow-up research in this direction remains limited to date.

**Reliable uncertainty estimates**   A distribution-free framework known as Conformal Prediction (CP) produces prediction sets that guarantee inclusion of the ground truth with a user-specified probability. By using an additional calibration set, CP transforms heuristic model confidence scores into rigorous uncertainty estimates. This technique is particularly renowned for being model-agnostic, simple, and highly flexible (Angelopoulos & Bates, 2021). Recently, CP has been applied to segmentation tasks (Wieslander et al., 2020; Brunekreef et al., 2024; Mossina et al., 2024; Wundram et al., 2024a), benefiting from these advantages and reflecting the growing interest in the framework for structured prediction. However, the validity of conformal prediction requires exchangeable draws for train, calibration, and test data. In practical settings test data often comes with distribution shift, which can coverage to fall below the nominal level and predictions can become overconfident. Such overconfidence yields prediction sets that are too tight, suppressing deferrals and increasing missed or undersegmented regions which can be especially detrimental in safety-critical tasks. We encourage further research to explore novel applications and benchmark CP against existing architectures.

**Generative models** Given the established time-lag in the adoption of SOTA techniques from the broader generative modeling field, there remains a persistent opportunity for research to effectively bridge the gap by translating the latest architectural and methodological improvements from image generation into more performant and robust segmentation solutions. For example, DDPMs were initially proposed for general image generation (Ho et al., 2020), but was quickly adopted for stochastic segmentation. Furthermore, flow-based models also enjoy many benefits yet to be explored in this field, such as exact likelihood modeling and faster sampling compared to DDPMs (Lipman et al., 2022).

### 7.9 Towards complex scene understanding

A clear trend shows that most uncertainty applications remain narrowly focused on binary semantic segmentation. While limited work exists in probabilistic instance segmentation (Kohl et al., 2019a; Morrison et al., 2019), there is a notable absence of research addressing uncertainty in this domain, as well as in panoptic segmentation. Furthermore, work involving uncertainty across multiple classes is also scarce. This represents a critical gap, given that contemporary segmentation research is increasingly centered on multiclass problems across all segmentation types (Kerssies et al., 2025), often even incorporating class hierarchies (Atigh et al., 2022). Consequently, the significance of research focused solely on binary-class semantic segmentation is diminishing.

## 8 Guidelines and recommendations

This section synthesizes the preceding discussions into practical guidelines and recommendations. We will first critically examine specific methods and offer criteria for selecting the optimal approach for different tasks. Subsequently, we will discuss the essential topic of evaluating uncertainty and the potential biases that can arise. The section concludes with a more philosophical look at what constitutes truly meaningful uncertainty in machine learning.

### 8.1 Method selection

The proposed framework is designed to simplify the complex landscape of available techniques by deconstructing the decision into a series of manageable questions. By systematically addressing these questions, practitioners can navigate the available options more effectively and clearly justify their final choice of method.

---

**Decision framework**:

1. **Baseline**: Does the vanilla backbone reach sufficient performance? (Section 8.1.1)
2. **Reducible**: Is the target uncertainty reducible? (Section 8.1.2)
3. **Purpose**: Is the uncertainty auxiliary or crucial to the main task? (Section 8.1.3)
4. **Data regime**: Are labels multi-annotated or multi-class? (Section 8.1.4)
5. **Budget**: Are there budgetary constraints that limit compute? (Table 7)

For each element, evaluate potential method with criteria:

- Prioritizing methods with stronger **evidence** in literature.
- Considering the **theoretical soundness**.
- Favoring **architecture-agnostic** and **post-hoc** designs.
- Evaluating **computational efficiency**, prioritizing methods that offer fast sampling time.

---

The final two points address the modern paradigm of using large, pre-trained models. These models are typically fine-tuned once for a specific task but are then deployed for continuous use. While the initial training is a significant one-time cost, deployment involves ongoing inference. Therefore, optimizing for

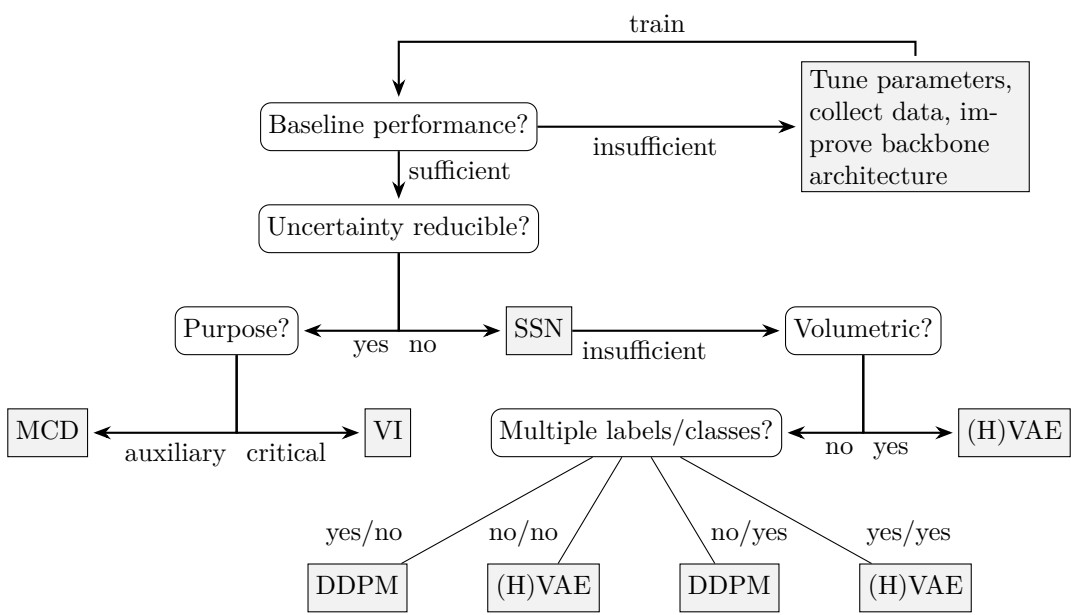

Figure 15: Flowchart illustrating the complete framework for selecting the appropriate uncertainty modeling method.

inference efficiency is paramount and will lead to substantial long-term savings in computational resources. A comprehensive comparison of available methods is provided in Table 7, which will be used as a key reference when making recommendations. The complete selection process for our proposed framework is outlined in the flowchart in Figure 15. We now proceed to detail each component of this framework in the subsequent sections.

### 8.1.1 Baseline

Probabilistic segmentation models perform best when they are built on a robust, capable backbone. Hence, it is crucial to establish a strong deterministic baseline first. If the underlying model underfits the data or fails to learn the task, adding a probabilistic layer will not solve this issue. Only when the baseline is solid can probabilistic modeling effectively provide calibrated uncertainty and support decision-making. Besides model backbone, improvements through data collection, parameter tuning, and different training schemes can also be explored.

### 8.1.2 Reducibility of uncertainty

Unless it is the core scientific contribution, we argue that disentangling uncertainty sources is often unnecessary for dense prediction applications. The primary benchmark for success is *improved performance* on a downstream task. If a model's total uncertainty score achieves this, its practical value is already established, regardless of our ability to precisely attribute that uncertainty to its underlying sources. This pragmatic view avoids the need for complex validation methods designed solely to prove disentanglement. Instead, the distinction between aleatoric and epistemic uncertainty is most valuable as a conceptual guide to communicate whether the uncertainty is reducible and, consequently, to select the most appropriate modeling strategy, as noted in prior work (Der Kiureghian & Ditlevsen, 2009; Faber, 2005).

Disentanglement shouldn't be mistaken for the primary objective, but might be a useful diagnostic tool. For example, it makes sense to *assume* disentanglement of the uncertainty to choose the appropriate method. In a theoretical view, a model that perfectly captures all learnable structure would have its remaining prediction error driven solely by irreducible noise. This suggests that the fitted parameters can be treated as near-point estimates, and consequently, most residual uncertainty would be expected to arise from the predictive distribution rather than from parameter variability. In contrast, if we were to model the reducible noise within

the parameter distribution by markedly narrowing the predictive distribution while the parameter variance remains materially above zero, we would be suggesting that a single sample from the parameter distribution could yield predictions with near-zero variance. This seems in tension with the premise of inherent data noise, since some uncertainty should still appear in predictions when the data-generating process is noisy. Thus, although the disentanglement of uncertainty is a theoretical debate (see Section 7.1), both our analyses and literature overview conclude that it remains a useful tool for method selection. Specifically, by modeling parameter distributions for reducible uncertainty and feature distributions for irreducible uncertainty. While a conventional neural network, by contrast, should theoretically be able to model both uncertainties within the feature-based output distribution, the empirical failure of this simple approach to provide robust uncertainty quantification is, in fact, the central driving force behind the entire field of study.

### 8.1.3 Purpose of modeling

Once a decision has been made on the uncertainty reducibility, the next step is to determine its purpose. For cases where the uncertainty is irreducible, our literature overview has identified a single use case: estimating Observer Variability (Section 6.1). For reducible uncertainty, this overview identified three key tasks. For the task of Model Generalization (Section 6.4), uncertainty is not the primary objective, but a beneficial byproduct of the modeling approach. In contrast, for Active Learning and Model Introspection (Section 6.2 and 6.3, respectively), uncertainty quantification is the core objective and must therefore be critically sound and theoretically justified.

**Observer Variability**  As demonstrated in Table 6, the top-performing methods for this task are SSN and DDPM-based approaches. Given its model-agnostic nature, fast sampling capabilities, compatibility with 3D data, and strong recent benchmarks (Kahl et al., 2024; Monteiro et al., 2020), we recommend prioritizing SSNs. Should the performance prove insufficient, further exploration based on specific data characteristics and computational constraints is warranted. As discussed in Section 7.6, the existing literature on the data dependency of these specific methods is limited. However, we will provide an overview and discussion in the following section based on the available studies.

**Model Generalization**  Though their theoretical grounding are limited (see discussion in Section 7.5, a pragmatic first step is to use either MC Dropout or TTA. Both methods are simple, inexpensive, and have been shown to be effective. Their limited theoretical support is not a major concern here, as the uncertainty is auxiliary to the main task of improving generalization. Nevertheless, MC dropout has more evidence in literature to perform well (see Table 4) and is generally better understood (i.e. the connection to variational inference made in Section 5.1) than TTA (its connection to Bayesian learning is under discussion as mentioned in Section 7.1).

**Active Learning & Model Introspection**  In contrast to Model Generalization, the reducible uncertainty is a crucial factor for both tasks. MC Dropout remains overwhelmingly the most popular uncertainty estimation method (see Tables 2 and 3) due primarily to its minimal implementation overhead, making it an easy baseline for numerous papers. Despite this prevalence, studies have reported that the simple random sampling baseline in Active Learning remains a highly competitive approach (Sinha et al., 2019). Recent benchmarks consistently show that ensemble methods are superior (Kahl et al., 2024), and Variational Inference (VI) closely trails ensembling while clearly outperforming MC Dropout(Ng et al., 2022). Since VI offers a more theoretically sound Bayesian approach that is also more cost-efficient than full ensembling, we recommend researchers adopt rigorous methods like VI for uncertainty characterization. Furthermore, caution is warranted concerning the aggregation methods detailed in Section 7.2. Namely, entropy calculation, often used to quantify uncertainty severity, is frequently severely overestimated. Future research must therefore pursue fundamentally sound methodologies that avoid these aggregation pitfalls.

### 8.1.4 Nature of the data

Our analysis in Section 7.6 indicates that uncertainty modeling is strongly data-dependent and remains largely inconclusive for modeling parameter distributions. Nonetheless, a limited but practical ordering can be established for modeling irreducible uncertainty via feature distribution methods, using criteria like class count, labels, and data dimensions.

**Multi-label, binary class**   The LIDC-IDRI benchmark heavily influences current literature, leading to a focus on (H)VAE-based methods. However, as shown in Table 6, DDPMs achieve superior scores on this key dataset. While (H)VAE approaches remain more popular for other datasets, the lack of robust benchmarking makes broader conclusions difficult. We therefore recommend DDPMs for superior performance, but recognize that (H)VAEs remain a viable alternative when constrained by computational limits.

**Single-label, binary class**   When evaluating single-label problems, accounting for ground-truth uncertainty (i.e., observer variability) is critical. The only rigorous verification method involves ablation studies on multi-annotated datasets where the model is trained only on a single mask. Since these crucial studies are performed almost exclusively on (H)VAE-based approaches (e.g., (Zhang et al., 2022d; Gao et al., 2022; Zepf et al., 2023a; Selvan et al., 2020), plus one instance with PixelSeg (Zhang et al., 2022c)), our recommendation favors (H)VAE models due to this strong accumulated evidence. However, given their superior mode coverage (Xiao et al., 2021), there is no indication that DDPMs would perform poorly in this specific ablation.

**Single label, multi-class**   Crucially, we have not found a multi-annotated, multi-class dataset used for ablation by training only on a single mask. In the absence of this rigorous evidence, we must draw inferences from related applications, such as Model Generalization. Within this context, Table 4 suggests that DDPMs are a popular choice.

**Multi-label, multi-class**   In the rare instances of multi-class, multi-annotated problems, such as the Cityscapes dataset when combined with artificial labels, (H)VAEs currently have the most supporting evidence (Kohl et al., 2019b; Gao et al., 2022).

**Volumetric data**   For problems involving 3D data, the primary evidence strongly supports the use of (H)VAE-based methods (Viviers et al., 2023c;a; Long et al., 2021a). This body of work encompasses both multi-class single-annotated and binary-class multi-annotated datasets. Given that this is the most substantial lead, outside of SSNs, we recommend the adoption of the (H)VAE model architecture for 3D uncertainty estimation.

## 8.2 Evaluation and reproducibility

Based on our literature overview and in-depth discussions, this section outlines general and task-specific recommendations aimed at improving evaluation and reproducibility. Compliance with these guidelines is essential for validating experimental findings and ensuring comparability across the broader research landscape.

### 8.2.1 General

**Reporting**   Achieving robust reproducibility and generalizability hinges on detailed reporting, an area where current research practice often falls short, as discussed in Section 7.3. Their inconsistent application and reporting inhibit the validation and critical assessment of published methods, eroding confidence in the field's reported progress. Furthermore, the routine omission of source code and model checkpoints exacerbates this issue, making result verification impractical. This practice stands in sharp contradiction to the standards of modern machine learning, which mandates standardized data splits, clear evaluation metrics, and the highly encouraged release of code and model weights.

Table 7: Comparison between methods. $N$ represents number of parallel or sequential models in ensembles and DDPM denoising, respectively. *Params.* indicate the number of parameters required for feature-based methods. For parameter-based modeling, this indicates how the parameter count scales with the uncertainty method. *Samples* indicate how many evaluations are needed for a single forward pass for feature-based modeling. For parameter-based modeling, this indicates if the uncertainty evaluation quality scales with the number of forward passes of parameter-based approaches.

| Method | Advantages | Limitations | Params. | Samples | Likelihoods | Extra notes |
|---|---|---|---|---|---|---|
| ———–FEATURE MODELING——– | | | | | | |
| SSN | Model agnostic, fast sampling | possibly unstable | ~41M | 1 | explicit, appr. | Occasional diagonal covariance during training. No prior, likelihoods impractical. |
| PixelCNN | High sample quality | Sequential sampling, weak global structure | ~3.4M | $H \times W$ | Explicit, exact | |
| CAR | Fast sampling, flexible | Unstable training, poorly defined objective, extensive tuning | ~43M | 1 | implicit | Additional losses for segmentation |
| (H)VAE | Fast sampling, flexible, interpretable latent space | Mode/posterior collapse, amortization gap | ~(19)5M | 1 | appr. lower bound | additional deep supervision, GECO objective |
| DDPM | Flexible, expressive | Sequential sampling | ~30M | $\times N$ | appr. lower bound | Compute can be reduced with DDIMs at cost of sample quality |
| ———–PARAMETER MODELING——– | | | | | | |
| LA | Model agnostic, no extra parameters | Local approx., full posterior infeasible | $\times 1$ | No | Explicit exact | Usually on last layer |
| Explicit VI | Full weight posterior; principled uncertainty | slow convergence | $\times 2$ | No | appr. lower bound | Usually on selected layers |
| Ensemble | simple, parallelization | expensive, member correlation risk | $\times N$ | Yes | Implicit | variants: M-Heads, MoE |
| MC dropout | Minimal code change; regularizes; model-agnostic | poor uncertainty, not principled | $\times 1$ | Yes | Implicit | variants: concrete, drop-weights, theoretical groundedness to be disputed |
| TTA | Zero training changes; boosts robustness | boundary errors, aggregation sensitivity | $\times 1$ | Yes | Implicit | |

**Structural splitting**   Furthermore, to prevent information leakage and ensure a truly generalizable model, data must be split strategically based on its underlying structure. For instance, splitting by patient, time point, or acquisition condition (e.g., device, institution, or environment). A notable failure to enforce the critical step of patient wise splitting is frequently observed (see, for example, (Baumgartner et al., 2019; Valiuddin et al., 2024b; Monteiro et al., 2020)).

**Inference details**   To ensure fair and accurate comparisons, the number of forward passes used to generate uncertainty maps and evaluation scores must be explicitly reported for every method. This is a prerequisite because the quality of the uncertainty estimate is often dependent on this hyperparameter (see Table 7). Such transparency, exemplified by the standardization work of Zbinden et al. (2023), allows the community to determine optimal validation strategies. Similarly, the specific parameters for any data augmentation must be detailed, particularly when using TTA (Section 5.3). For latent variable models, it is crucial to specify the dimensionality of the latent space and detail the precise mechanism for incorporating these variables into the network architecture (e.g., through tiling in Kohl et al. (2019b) or by using scale and translation parameters as done in Kassapis et al. (2021)).

**Quantifying uncertainty types**   As argued by Kahl et al. (2024), claiming to quantify an uncertainty type would ideally require validation across *all* relevant applications. For example, validation of epistemic uncertainty requires a clear distribution shift in the data (domain, semantic, covariate etc.), but it is often found in literature that either no distribution shift or the required metrics are used (Zhang et al., 2022a; Wang et al., 2019a; Postels et al., 2019; Mukhoti & Gal, 2018; Mobiny et al., 2021; Whitbread & Jenkinson, 2022). As discussed in Section 6.1, aleatoric uncertainty should be validated with multiple labels and appropriate metrics, but can often be found to be evaluated on a single annotator or without the use of distribution-level statistics (Wang et al., 2019a; Whitbread & Jenkinson, 2022; Kendall & Gal, 2017; Liu et al., 2022b; Schmidt et al., 2023; Savadikar et al., 2021). Hence, simply avoiding any serious claims related to either type alleviates this burden, and allows one to focus solely on the task at hand. In contrast to the work of Kahl et al. (2024), however, we do not consider calibration of models itself as a downstream task, but rather as a tool to gauge model reliability to be later used for other tasks. As a consequence, the pixel-wise uncertainties can be later used for possible downstream applications.

**Aggregation**   The aggregation method should be considered as a standalone design choice (discussed in Section 7.2) and researchers must explicitly state how per-pixel probabilities are combined. Ideally, multiple aggregation techniques should be evaluated to rule out potential biases, such as those related to object size. When working with volumetric data, it is crucial to evaluate both 3D volumetric and 2D slice-based approaches. Furthermore, entropy must be calculated in one of two ways: either correctly, using the true likelihood under the model or by using a factorized assumption. If the latter approach is taken, this significant methodological limitation must be made explicit.

**Feature extractors**   The feature extracting backbone is a key component in uncertainty modeling (Section 7.7). Thus, any proposed method or improvement must be ablated across diverse feature extractors to confirm its general functionality. For instance, latent variable methods should demonstrate performance stability when interchanging CNN or ViT encoders, and parameter modeling approaches should similarly be tested across various architectures.

**Dataset**   Data dependency is a crucial element in uncertainty modeling (Section 7.6). For new methodological contributions, ablation across diverse datasets is essential. These datasets should encompass variations in: number of classes, modality, input channels, dimensionality (2D vs. 3D), and annotation types (single vs. multi-annotated). This scope should also extend to complex tasks beyond simple semantic segmentation, notably instance and panoptic segmentation.

### 8.2.2 Task-specific guidelines

**Observer Variability**  We recommend the standard of 16 predictive samples for evaluating the GED and HM-IoU on the LIDC-IDRI dataset, a convention established in literature (Table 6). Another commonly used dataset is the CityScapes benchmark. for this, it is advisable to use the label-switching parameters proposed by Kohl et al. (2019b). Other multi-annotated datasets, such as those from the QUBIQ Challenge (Menze et al., 2021), include MRI and CT scans of various organs (Valiuddin et al., 2024b; Ji et al., 2021), although they follow less standardized preprocessing. The RIGA dataset for retinal fundus images (Almazroa et al., 2018) can be used as in Wu et al. (2022). Furthermore, it is crucial to explicitly report the strategy for handling correct empty segmentation maps; we recommend assigning a maximum score (Valiuddin et al., 2024b) rather than removing NaNs (Kohl et al., 2019a). Finally, evaluations should include an ablation study on the number of annotations to fully assess the generalization capabilities of a model.

**Active Learning**  Researchers should report the relative change in performance (corrected for random sampling as shown in Equation (37)) and evaluate sample diversity using metrics like FID. Furthermore, an ablation study should be conducted where varying percentages of the data are withheld from the initial training pool to assess performance under different data availability scenarios.

**Model Introspection**  It is essential to report PAvPU and CoV (see Section 6.3) based on rigorous uncertainty calculations. Furthermore, studies must critically analyze the relationship between the characteristics of the ground truth segmentation mask and the model's predicted error. This analysis is crucial for ensuring that the model's uncertainty estimates are reliable and not biased by specific data attributes (i.e. size, complexity, location, contrast).

**Model Generalization**  Report the relative change in performance as discussed in Section 6.4.

### 8.3 Limitations in the recommendations

**Selection bias**  A limitation of this review stems from the prevalence of specific methods in the current literature such as VAE-based models. While dominance of such methods may reflect the technique's genuine efficacy or versatility, it may also be attributed to factors such as ease of implementation, readily available codebases, or simplicity of initial adoption, thereby potentially introducing a selection bias into the overall analysis. Furthermore, their extensive application may not inherently confirm their superiority. In fact, it could simply reflect the community momentum and historical popularity the technique.

**Literature gaps**  The heterogeneity and lack of rigor in the current literature necessitated a degree of generalization in method classification, which is a key constraint on our comparative analysis. To maintain clarity, we primarily refer to methods by their underlying conceptual class (e.g., VAE, DDPM) rather than specific implementations, as many variants (including fine-grained hierarchical or categorical versions) offer only incremental changes. Exceptions are made solely for highly task-specific methods, such as SSNs and GAN-based CAR, which lack a broader architectural category. This constraint was amplified by the widespread absence of standardized benchmarks and comprehensive failure analyses in literature. Consequently, to ensure analytical coherence across all surveyed tasks, we sometimes infer the properties of a specific method from its parent class or rely on qualitative observations based on general architectural principles, prioritizing broad insight across fragmented literature over implementation-specific empirical precision.

**Soundness vs. prevalence**  Finally, a notable tension emerged between theoretical soundness and empirical prevalence. Methods built upon stronger theoretical justifications, such as those utilizing principled Bayesian inference, are often less prevalent in the literature than more heuristic, theoretically weaker, approaches such as MC dropout or ensembling. This disparity suggests a practical barrier to adoption (e.g., complexity or computational cost). In several instances, our analytical preference for theoretical rigor superseded the volume of available literature on more widely adopted alternatives. This choice risks under-representing the pragmatic, empirically dominant approaches that currently drive much of the applied progress in the field.

### 8.4 Criteria for good uncertainty

Following the review of the fundamental need and methodological approaches for uncertainty modeling, this analysis culminates in the essential assessment of utility. This section transcends evaluations based on algorithmic complexity, theoretical soundness, and standard metrics to articulate the defining characteristics of "good" uncertainty. We will therefore establish the practical criteria required to ensure the resulting estimates possess genuine relevance in real-world scenarios.

#### 8.4.1 Reliable

Trustworthy systems fundamentally require high accuracy in the point prediction, as being correct most of the time is the prerequisite for the model's uncertainty to be low (sharp). Secondly, and critically, the model must be well-calibrated. In cases of potential inaccuracy, the model must reliably report high uncertainty to maintain trustworthiness. A miscalibrated model, often manifesting as dangerous overconfidence, is fundamentally unreliable. High confidence misleads users to assume low risk, which is particularly problematic in safety-critical fields like medicine or autonomous driving, thus compromising safe decision-making.

#### 8.4.2 Explainable

To be effective, uncertainty must be well-communicated and easily understood. For example, a raw entropy map is often insufficient because it provides only a relative comparison between pixels while lacking absolute, actionable meaning. Therefore, a better approach is to process this raw data into an interpretable signal for the end-user. This can be achieved by displaying distinct, plausible alternative boundaries or by using a bright color to highlight only the regions of highest uncertainty, which effectively guides an expert's attention where it is needed most.

#### 8.4.3 Actionable

The key insight is that uncertainty should not be seen merely as a measure of model doubt, but as a crucial actionable signal. In practical applications, this signal is used to trigger specific, risk-mitigating behaviors, such as altering a plan, halting a procedure, or deferring control to a human expert, to ensure safety and prevent errors. For effective utility, users must be clearly informed about the reducibility of the uncertainty. High uncertainty should signal whether there is a need to collect more training data or refine the model or if it simply represents irreducible, inherent data noise. In this latter case of irreducible uncertainty, the system must still recommend a specific risk-mitigating strategy based on the bounds of the prediction.

#### 8.4.4 Unbiased

A fundamental tenet of robust and equitable modeling is that the system's predictions must remain unbiased with respect to the underlying data distribution. To address this core challenge, we will categorize and discuss several common type of biases, providing examples of how they can manifest.

**Distributional bias** If the validation data does not reflect the real-world deployment environment, the model's uncertainty estimates will not be reliable in practice. This bias can stem from subtle shifts in acquisition conditions or population characteristics. Therefore, to ensure robustness, datasets must represent a balanced collection encompassing diverse acquisition and demographic variations. Examples include an object detection model validated exclusively on footage from sunny highways deployed in a snowy snowstorm. Or, a diagnostic model validated on MRI from one hospital and specific demographic deployed at a new institution with different scanners and a diverse patient population.

**Annotation bias** Relying on a single ground-truth mask introduces a significant source of bias, as the model learns to be certain about one annotator's idiosyncratic choices, even where boundaries are ambiguous to other experts. This results in a false sense of certainty, where the model's confidence reflects its ability to replicate a single opinion rather than the inherent ambiguity of the task. Furthermore, the selection of annotators is a critical consideration. To mitigate biases from individual annotators and their tools,

researchers should collect multiple annotations from a diverse group of experts to build a ground truth that more accurately captures the inherent ambiguity of the task. Examples of this can be delineations provided by highly trained medical experts often differing systematically from those of trainees or multiple experts, which may have their own valid interpretation of the boundary. Another example of this can be that a polygon tool for annotating encourages simplified masks with straight edges, while a freehand brush tool allows for more detailed curves but can introduce minor inconsistencies.

**Modeling bias**  Different uncertainty quantification methods may have their own intrinsic biases. For instance, Gaussian densities assume single-moded uncertainty as well as a light-tail bias. Consider segmenting a brain tumor with ambiguous boundaries, a single-moded Bayesian model produces only one average segmentation. This misrepresents the true, multimodal uncertainty to a clinician. For autonomous vehicles segmenting a road, the light-tail bias, often inherent in models relying on assumptions like the Gaussian distribution, causes the system to view critical safety hazards, such as massive potholes, as statistically near-impossible events because training focused primarily on smooth surfaces.

**Sampling bias**  A system trained to reduce uncertainty often focuses its labeling budget on rare or complex outlier samples. This wastes resources by developing expertise in statistically insignificant cases, ultimately resulting in a model that fails to reliably generalize or improve performance in the common, high-value scenarios it was designed to analyze. For example, in flawed active perception, a soccer robot aiming to segment the ball focuses its attention where segmentation uncertainty is highest, i.e. the visually complex sidelines. This draws focus away from the simple green field, causing the robot to become an expert at tracking the ball in the least important part of the game. Another example can be a model used in satellite imaging may focus its uncertainty-driven budget on unique, complex buildings instead of common houses resulting in a model expert in architectural oddities but failing to reliably segment critical residential areas.

**Validation bias**  Overemphasizing metric improvement can incentivize models to undermining the original goal. A textbook case of Goodhart's law: "*When a measure becomes a target, it ceases to be a good measure*". Examples of this are optimization of metrics that minimize the discrepancy between the prediction and ground-truth distribution can incentivize the model to simply replicate the observed ground-truth masks rather than capture the full range of plausible, unseen segmentations. Furthermore, a model can achieve a perfect calibration (ECE score) while remaining dangerously useless for quantification; this occurs when the model only expresses binary confidence (100% or 0%). While technically calibrated, this failure mode is useless because the system fails to communicate the necessary nuanced doubt about ambiguous regions. Another example can be entropy-guided uncertainty can perversely incentivize models to suppress doubt and minimize uncertainty in genuinely ambiguous regions to achieve better metric scores.

---

**Key takeaways**: Good predictive uncertainty is defined by:

- **Reliability**: The uncertainty should accurately reflect the model's true error rate.

- **Explainability**: The source and meaning of the uncertainty should be clear to the end-user.

- **Actionability**: The uncertainty estimate must be useful for downstream decision-making.

- **Unbiasedness**: The estimates should not be systematically skewed by factors like subgroup characteristics.

---

# 9 Conclusion

This review synthesizes the fragmented literature on uncertainty in semantic segmentation, establishing a unified framework to bridge the gap between method developers, task specialists and applied researchers. Our analysis highlights critical challenges, including the nuanced distinction between uncertainty types, flawed assumptions in spatial aggregation, and a lack of standardization. We identify active learning, data-driven benchmarks, and transformer-based models as the most promising avenues for future research, particularly for extending novel uncertainty methods to holistic tasks like instance/panoptic segmentation. Our conclusion provides a clear method selection guide and places a strong emphasis on the need for future work to produce uncertainty that is reliable, explainable, actionable, and unbiased.

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
