# OpenReview forum: "A Review of Bayesian Uncertainty Quantification in Deep Probabilistic Image Segmentation"
_TMLR — Accepted by TMLR_

### Review · Reviewer_rS4H · 2025-09-08

**Summary Of Contributions:**

The authors review common approaches to probabilistic image segmentation, covering both aleatoric uncertainty modelling techniques (e.g. based on GANs, VAEs and diffusion models) and methods for estimating epistemic uncertainty (mainly covering MC-dropout and ensembling).
***
***
***
***







Strengths:
- The studied problem is interesting/important, a review of this topic could definitely be useful.
- The paper is well-written overall, clean figures etc, solid work overall.
***




Weaknesses:
- At least to me, the paper feels somewhat disjointed overall. The different sections and subsections don't quite tie together in an intuitive way. For example, the transition from Section 3 to Section 4 could be smoother, and it could be made more clear at the start of Section 4 what will be covered in this section, how it relates to the rest of the paper etc. In general, it could be made more clear what is covered in the different sections/subsections and _why_ this is covered (this is a long paper, and I quite often found myself reading and thinking roughly _"yes, this is a solid description of method X, but why am I reading this? Will this be used later in the paper in some way? If so, how? And how did we get here? How does this relate to the previous sections and what the paper aims to do overall?"_).
- Personally, I didn't find this review overly interesting/useful, I don't quite feel like it provided me with a lot of new knowledge or important new insights. At the same time, I'm not entirely sure that someone who is less familiar with uncertainty estimation methods would find this to be a good introduction to the field, I think there are more pedagogical / digestible / easy-to-follow resources for that. In short, it's somewhat unclear to me what the main target audience for this paper is, or what type of person who would find this paper really helpful/useful/interesting.
***




Questions/suggestions:
- I don't quite understand Figure 11, don't quite see what this is intended to illustrate/explain?
- In Section 7.6, _"Table 1 shows AB (Chen et al., 2022) performs almost identically to CCDM (Zbinden et al., 2023)"_: I don't quite understand how this is shown in Table 1? Or, what exactly do you mean here?
- Section 7.7, "Reliable uncertainty estimates" paragraph: Should at least mention that the coverage guarantee of conformal prediction depends on the assumption of exchangeably drawn train/calibration and test data (true for i.i.d. data, for example)? At least in my opinion, this is a quite serious limitation of CP, it can produce overconfident predictions in many practical settings due to distribution shifts (when the exchangeability assumption generally does not hold) etc.
***





Minor things:
- Section 2, "Ronneberger Ronneberger et al. (2015) introduced the U-Net": typo.
- Section 3: "the predictive distribution from a new datapoint" --> "the predictive distribution for a new datapoint"?
- Section 3, "Hence, uncertainties stemming from the conditional likelihood distribution are classified as either aleatoric, implying from the statistical diversity in the data, or epistemic, which stems from the posterior, i.e. the variance of the model parameters": perhaps just restructure a bit?
- Section 3.1: "does not include any prior knowledge on the structure of the parameter distribution but can be achieved" --> "does not include any prior knowledge on the structure of the parameter distribution, but this can be achieved" or similar perhaps?
- Section 3.1: "we make use of function" --> "we make use of a function"?
- Section 4.1.1, "true probabilities in modern neural networks Guo et al. (2017)": citation formatting, "Guo et al. (2017)" --> "(Guo et al., 2017)".
- 4.1.1, "considered.Additionally": typo.
- 4.1.1, "imbalance Jungo et al. (2020); Neumann et al. (2018)": citation formatting.
- 4.2: "to instead learn the approximate through" --> "to instead learn the approximation through"?
- 4.2, "optimization trajectories Zheng et al. (2022)": citation formatting.
- Fig 7 caption, "network Kassapis et al. (2021)": citation formatting.
- 4.2.2, "Sinkhorn divergence Cuturi (2013)": citation formatting.
- 5, "Mean-Field Variational Inference Blundell et al. (2015), Markov Chain MonteCarlo (MCMC) Neal (2012), Monte-Carlo Dropout Gal & Ghahramani (2016); Kingma et al. (2015), ModelEnsembling Lakshminarayanan et al. (2017), Laplace approximations Mackay (1992), Stochastic GradientMCMCs Korattikara Balan et al. (2015); Springenberg et al. (2016); Welling & Teh (2011), assumed densityfiltering Hernández-Lobato & Adams (2015) and expectation propagation Hasenclever et al. (2017); Louizos& Welling (2016)": citation formatting.
- 5, "Test-time augmentation, often mistakenly regarded as an aleatoric method, is addressed in Section 5.3. Finally, Laplace Approximations are examined in Section 5.2": incorrect ordering.
-  5.1.1: "The relationship between p and the magnitude of the model weights has also be exploited" --> "The relationship between p and the magnitude of the model weights has also been exploited" / "The relationship between p and the magnitude of the model weights can also be exploited"?
- 5.1.1, "prune neural networks Gonzalez-Carabarin et al. (2022)": citation formatting.
- 5.1.1: "where the weights of the network, rather that its" --> "where the weights of the network, rather than its"?
- 5.2, "post-hoc to .e.g.": typo.
- 5.2: "by treating neural network partly probabilistic" --> "by treating the neural network partly probabilistic" / "by treating neural networks partly probabilistic"?
- 6.1: "each annotator is consistent in his delineation" --> "each annotator is consistent in their delineation"?
- 6.1, "inductive biases Kohl et al. (2019a); Zepf et al. (2024)": citation.
- 6.1: "it can also be possible to modeling the annotator distribution" --> "it can also be possible to model the annotator distribution"?
- 6.1, "labeling styles of annotators Zepf et al. (2023a)": citation.
- Missing period at the end of the caption for Table 2 and 3.
- 6.2, "predictive entropy Kasarla et al. (2019); Shen et al. (2021); Xie et al. (2022) or variance Ozdemir et al. (2021)": citation.
- 6.2, "Yang et al. (2017); Sourati et al. (2018); Kasarla et al. (2019); Mahapatra et al. (2018); Sinha et al. (2019); Siddiqui et al. (2020); Li & Alstrøm (2020); Kim et al. (2021); Shen et al. (2021); Burmeister et al. (2022); Ma et al. (2024b)": citation.
- 6.2: "literature use MC dropout of ensembles" --> "literature uses MC dropout or ensembles"?
- End of 6.2, "Ozdemir et al. (2021); Shen et al. (2021); Wu et al. (2021); Mahapatra et al. (2018); Sinha et al. (2019)" and "Gaillochet et al. (2023); Wu et al. (2021); Kasarla et al. (2019)": citation.
- 6.3: "Hence, Roy et al. (2018b) propose to use the Coefficient of Variation (CoV) addresses this by" --> "Hence, Roy et al. (2018b) propose to use the Coefficient of Variation (CoV) to address this by"?
- 6.4, "Saha et al. (2021a); Wolleb et al. (2021); Wu et al. (2023b); Amit et al. (2021); Bogensperger et al. (2023); Zbinden et al. (2023); Viviers et al. (2023a); Zepf et al. (2023a); Hu et al. (2022)": citation.
- 7.1.1, "total uncertainty Depeweg et al. (2018)" and "residual risk Apostolakis (1990); Helton (1997)": citation.
- 7.1.1: "This is dependency is also demonstrated" --> "This dependency is also demonstrated".
- 7.1.1, "the data Kahl et al. (2024)", "quantification methods de Jong et al. (2024); Mucsányi et al. (2024)", "distributions Kingma et al. (2015)", "Zhang et al. (2022c); Monteiro et al. (2020); Valiuddin et al. (2024b); Gao et al. (2022)", "Kendall et al. (2016); Roy et al. (2018b); DeVries & Taylor (2018); Mehrtash et al. (2020); Burmeister et al. (2022); Iwamoto et al. (2021)": citation.
- 7.1.2: "but can often found to be evaluated on" --> "but can often be found to be evaluated on"?
- 7.1.2, "Wang et al. (2019a); Whitbread & Jenkinson (2022); Kendall & Gal (2017); Liu et al. (2022b); Schmidt et al. (2023); Savadikar et al. (2021)": citation.
- 7.2: "we further dive into import details and challenges" --> "we further dive into important details and challenges"?
- 7.2, "Mukhoti & Gal (2018); Camarasa et al. (2021); Bhat et al. (2022b); Mukhoti et al. (2020); Ma et al. (2024b); Shen et al. (2021); Li & Alstrøm (2020); Dechesne et al. (2021); Huang et al. (2018); Nair et al. (2020); Zepf et al. (2023b)", "redundant Shen et al. (2021); Burmeister et al. (2022)", "direction Liu et al. (2025); Burmeister et al. (2022); Viviers et al. (2023b)": citation.
- 7.2: "appropriately compare 2D- to 3D models" --> "appropriately compare 2D to 3D models"?
- 7.2: Looks like there's an extra space at the beginning of the Goodhart’s law quote.
- 7.4: "the notion that uncertainty methods work poor on the initial samples" --> "the notion that uncertainty methods work poorly on the initial samples"?
- 7.4: Multiple instances of incorrect citation formatting here as well. Same also for 7.6 (including in the Figure 13 caption). And for 7.7.
- 7.5, "enhance performance Chen et al. (2023b)": citation.
- 7.6: "while classified PixelCNNs and SSNs are classified as pixel-level methods, both can also be phrased as a latent-variable model" --> "while PixelCNNs and SSNs are classified as pixel-level methods, both can also be phrased as latent-variable models"?
- 7.7: "However, it’s debatable whether" --> "However, it is debatable whether"?
- 7.7: "In particular, such study can" --> "In particular, such a study can"?
- 7.7: "after training with SoftMax predictive distribution" --> "after training with a SoftMax predictive distribution"?

**Additional Comments:**

No additional comments.

**Audience:**

Yes

**Audience Explanation:**

The studied problem is interesting/important, a review of this topic could definitely be useful.

**Broader Impact Concerns:**

No concerns.

**Claims And Evidence:**

No

**Claims Explanation:**

The authors claim that _"By the end of this paper, readers will have a clear understanding of the various forms of uncertainty, their relevance to segmentation tasks, and a comprehensive grasp of the key challenges and open research directions in the field"_. Overall, I'm not entirely sure that the paper fulfills this claim, see "Weaknesses" above.

**Requested Changes:**

This is a quite well-written and solid paper overall, and I think it could be relevant for the TMLR audience.

However, I think the current version requires some clarifications and tweaks, see "Weaknesses" and "Questions/suggestions" above.

---

> ### Author Response · Authors · 2025-10-25
> **Response to review**
>
> We sincerely appreciate the reviewer's positive assessment. We agree that the current version will benefit from the suggested tweaks and clarifications and have addressed each point individually in the following sections of the rebuttal. We are confident these changes will further strengthen the manuscript's clarity and precision.
>
> **At least to me, the paper feels somewhat disjointed overall...**
>
> We agree that a clearer narrative path would strengthen the manuscript. To address this, we have implemented several revisions throughout the paper To create a more intuitive flow from one section to the next, we have added guiding text at critical junctures:
>
> 1. *Section 3.2: "Review Organization and Structure"*, which serves as a roadmap for the upcoming sections, explicitly addressing the transition from Section 3 to 4.
> 2. Revised introductory paragraphs for *Sections 4, 5, 6, and 7* to clearly state the purpose of each section, what it will cover, and how it connects to the paper's overall objectives.
> 3. Strategic summaries and key takeaways to ensure the reader remains oriented and understands the purpose of each discussion. These are distinguished with beige boxes.
>    - Key takeaways the end of *Section 3.1, 4.1, 4.2 and 5, 6, 8*.
>    - Summarizing bullet points in *Section 3.2, 6, 7 and 8.*
>    - A new decision framework in *Figure 15*.
>
> We are confident that these additions make the manuscript more cohesive and directly address the reviewer's concern. The reader is now regularly reminded of why something is being discussed and how it fits into the broader context of the paper.
>
> **Personally, I didn't quite feel like it provided me with a lot of new knowledge ...**
>
> By design, the paper does not fit the mold of a standard introductory survey in a single niche. Our paper is not only a deep-dive, but is also intended for researchers who are familiar (or even experts) in one sub-domain (e.g., methodology development) but seek to understand its connection to others (e.g., specific tasks or application areas), and vice versa. Hence, a key contribution is the synthesis of this fragmented field. We realize we must make this unique positioning clearer to the reader from the outset. To make this explicit, we have modified the Introduction to clearly frame the paper's objective.
>
> > "...Hence, this review is not only a conventional "zero-to-hero" survey; rather, ..."
>
> This framing clarifies that the paper's value lies not only in introducing uncertainty modeling concepts, but also in forging new connections and providing a unified perspective on a fragmented landscape.
>
> **I don't quite understand Figure 11, don't quite see what this is intended to illustrate/explain?**
>
> The goal of *Figure 11* is to provide an intuitive, visual comparison of the three dominant strategies for modeling parameter uncertainty in convolutional neural networks, showing how each approach "samples" a different convolutional kernel. To better communicate this core idea, we have revised the caption of *Figure 11* to be more explicit and descriptive. We also adapted *Figure 11a* to indicate more clearly that the two channels represent the mean and variance of the convolutional kernel distribution.
>
> **In Section 7.6, "Table 1 shows AB (Chen et al., 2022) performs almost identically to CCDM (Zbinden et al., 2023)": I don't quite understand how this is shown in Table 1? Or, what exactly do you mean here?**
>
> This was a typo and should have referred to *Table 6*. We have corrected this in the revised manuscript.
>
> **Section 7.7, "Reliable uncertainty estimates" paragraph: Should at least mention ...**
>
> The exchangeability assumption is a serious limitation of standard conformal prediction, and our discussion was indeed incomplete without mentioning it. We added this into the relevant paragraph (now in *Section 7.8*) to provide a more balanced and critical perspective.
>
> > "However, the validity of conformal prediction ..."
>
> **The authors claim that ...**
>
> We acknowledge the reviewer's concern and agree that the original statement of scope was indeed strong. To address this, we have removed the definitive claim from the introduction and have reframed the purpose of the review.
>
> > "Our aim is to provide a cohesive framework .."
>
> **Minor things**
>
> We thank the reviewer for their careful proofreading. We have corrected all identified typos and grammatical errors throughout the document.

---

> > ### Comment · Reviewer_rS4H · 2025-11-10
> >
> > Thank you for the response!
> >
> > I have read the other reviews and all author responses.
> >
> > The authors have provided a thorough rebuttal and made significant updates to the paper, they definitely seem to have put in a lot of effort into improving the paper based on the reviewer comments.
> >
> > I think this is a solid paper that could be relevant for the TMLR audience, and will thus recommend accept.

---

> > > ### Author Response · Authors · 2025-11-18
> > > **Reply to acceptance**
> > >
> > > We are very glad to hear that the revisions have successfully addressed your concerns and strengthened the paper. We also appreciate your time in reviewing the full discussion and for your recommendation to accept.
> > >
> > > Kind regards,
> > >
> > > The Authors.

---

### Review · Reviewer_HJet · 2025-09-10

**Summary Of Contributions:**

This review paper provides a survey of Bayesian uncertainty quantification in deep probabilistic segmentation, organizing a fragmented literature across domains like medical imaging and autonomous driving. It offers taxonomic organization of uncertainty modeling approaches and identifies practical challenges in the field.

**Strengths:** Comprehensive literature coverage; identifies important practical challenges; valuable as reference work for the field

**Weaknesses:** Perpetuates problematic epistemic/aleatoric framework despite acknowledging its limitations; lacks critical analysis of method effectiveness; insufficient empirical support for comparative claims; shallow treatment of validation approaches; limited practical guidance for method selection

**Additional Comments:**

Consider whether the goal is purely organizational (in which case acknowledge theoretical limitations more explicitly) or whether you aim to contribute to resolving conceptual issues (in which case provide deeper critical analysis and potential solutions).

**Audience:**

Yes

**Audience Explanation:**

Uncertainty quantification in segmentation is highly relevant to the TMLR community, particularly given applications in safety-critical domains like medical imaging and autonomous driving.

**Claims And Evidence:**

No

**Claims Explanation:**

* The paper repeatedly discusses uncertainty "disentanglement" methods while acknowledging these categories are "entangled or conflicting." No evidence is provided that existing methods can reliably achieve this disentanglement, making discussions of epistemic vs. aleatoric methods potentially misleading.

* The paper suggests methods addressing spatial coherence solve
  fundamental problems without providing evidence that these approaches
  actually improve uncertainty calibration.

* This evidence gap is particularly problematic given the paper's survey nature. Readers expect claims about method effectiveness to be supported by rigorous analysis of the literature, not assertions based on selective citation.

**Requested Changes:**

**Critical:**

* Either defend the epistemic/aleatoric distinction with new evidence and arguments, or reorganize around empirically grounded categories (e.g., by computational approach, application domain, or uncertainty estimation strategy).
* Replace anecdotal effectiveness claims with structured analysis using consistent evaluation criteria across methods. Include quantitative comparisons where possible rather than qualitative descriptions.
* Develop frameworks for evaluating uncertainty quality in segmentation
  contexts, especially for applications with ambiguous ground truth.
  Cooment on fundamental questions about what constitutes good
  uncertainty estimates. Poorly calibrated uncertainty estimates in medical imaging could lead to missed diagnoses or unnecessary procedures, while unreliable uncertainty in autonomous driving could affect collision avoidance systems. The authors should discuss validation strategies and potential biases in uncertainty estimates for high-stakes applications.
* Identify when and why specific methods fail, rather than only describing successes. Analyze patterns across applications to provide clearer guidance on practical trade-offs.
* Provide specific recommendations for method
  selection based on data characteristics, computational constraints,
  and application requirements rather than general observations.

**Strengthening:**

* Provide deeper analysis of spatial coherence implications: Section 7.2 identifies spatial aggregation as a key challenge, but the theoretical implications for uncertainty modeling frameworks deserve more thorough exploration.

---

> ### Author Response · Authors · 2025-10-25
> **Response to review**
>
> We thank the reviewer for their supportive and positive feedback. Guided by their suggestions, we have substantially revised the manuscript. The most significant changes are the restructuring of the manuscript around the empirical method of modeling uncertainty instead of the uncertainty type, and addition of an entirely new capstone section, *Section 8: "Recommendations and guidelines"*. This section was written to holistically address the reviewer's key concerns. Below, we provide a detailed, point-by-point response to each of the reviewers' comments.
>
> **Either defend the epistemic/aleatoric distinction....**
>
> We have fundamentally restructured the manuscript around the modeling strategy for uncertainty estimation. We have systematically de-emphasized the aleatoric/epistemic dichotomy throughout the manuscript, removing most direct references and retaining them only where essential for contextual understanding. To acknowledge the existing discourse while maintaining our new focus on modeling strategies, we have consolidated the discussion of epistemic and aleatoric concepts into a single, dedicated subsection (*Section 7.1*).
>
> **Replace anecdotal effectiveness claims...**
>
> We have worked to replace qualitative descriptions with quantitative data wherever possible. A key example is the expansion of *Table 7*, which now provides a structured comparison of all surveyed methods across concrete metrics, including parameter count, its scaling with uncertainty methods, whether the quality depends on the number of samples taken, and the number of evaluations required. We have also incorporated more direct comparisons for method selection (Section 8) based on the available evidence in the literature.
>
> A perfectly consistent, side-by-side empirical comparison is challenging. Our review revealed that this is largely due to a lack of standardized benchmarks and rigorous reporting in the existing literature. To address this directly and transparently, we have added a limitations *Section 8.3 " Limitations in the recommendations"*. This section explicitly informs the reader about the necessary generalizations we had to make.
>
> We hope the reviewer agrees that this is a responsible and thorough way to address their feedback.
>
> **Develop frameworks...**
>
> The reviewer rightly points out the need for a more explicit framework for evaluating uncertainty. While our initial submission introduced relevant metrics (*Section 6.1-6.4*) and recommendations (*Section 7*), we agree that this discussion lacked a central, cohesive structure. The feedback was the catalyst for us to synthesize and substantially expand on these ideas. To this end, we have introduced a dedicated subsection, *Section 8.2: "Evaluation and Reproducibility"*. This new section now offers a unified guide, recommending concrete, actionable practices for the field.
>
> **Comment on fundamental questions...**
>
> To address this, we have introduced a new dedicated section, *Section 8.4: "Criteria for good uncertainty".* This section provides a framework for evaluating uncertainty estimates by outlining four key criteria: reliability, explainability, actionability, unbiasedness.
>
> **Identify when and why specific methods fail...**
>
> While our original manuscript touched on these issues in several sections, we acknowledge that this analysis needed to be more prominent. Hence, we have now synthesized these points and more into the the dedicated *Section 7.5: " Method suboptimality"* . Furthermore, the new section *Section 8.1: "Method selection"* utilizes the discussion section to analyze across applications and provides guidance on model selection and discusses tradeoffs with help of the new *Table 7*, which holistically considers all modeling methods.
>
> **Provide specific recommendations...**
>
> We have introduced the new recommendation *Section 8.1*, that explicitly analyzes the practical trade-offs in method selection. This section systematically guides the reader by evaluating methods against key decision criteria. This analysis also provides two practical aids:
>
> 1. A decision flowchart (*Figure 15*) to guide the selection process
> 2. A comprehensive summary of trade-offs in the expanded *Table 7*.
>
> **Provide deeper analysis of spatial coherence...**
>
> We have revised and extended *Section 7.2*. We now include a concrete example with the help of *Figure 13*, to illustrate the critical importance of spatial coherence in uncertainty estimation. We believe this addition makes the theoretical implications tangible and directly addresses the reviewer's suggestion.
>
> **Consider whether the goal is purely organizational or...**
>
> Our principal goal is organizational: to provide a structured and comprehensive survey of the field. Following the reviewer's advice, we restructured the paper around modeling strategies rather than the conceptual disentanglement of uncertainty, which is no longer the central theme.

---

> > ### Comment · Reviewer_HJet · 2025-11-10
> > **Thanks**
> >
> > Thanks to the authors for making a substantial revision. It would have been much easier for me if TMLR required a diff file to be provided in TMLR. Nevertheless, the restructuring around modeling strategies, the addition of Section 8 with its decision flowchart (Figure 15), and the detailed failure analysis in Section 7.5 represent improvements that address my main concerns. I particularly appreciate the honest acknowledgment of limitations in Section 8.3 and the failure analysis with specific mechanisms and citations.
> >
> > While Table 7 does not provide the quantitative comparisons I initially envisioned, I recognize that this is a limitation of the field's current state rather than the authors' effort. The transparency about this issue is appropriate.
> >
> > The limitations currently discussed in Section 8.3 regarding the lack of standardized benchmarks and comprehensive empirical validation should be clearly stated in the paper's abstract and/or introduction. Readers should understand from the outset that method comparisons are based on available evidence rather than quantitative validation across standardized benchmarks. This transparency is essential for properly contextualizing the recommendations in Section 8.
> >
> > Subject to this revision, I recommend acceptance.

---

> > > ### Author Response · Authors · 2025-11-18
> > > **Minor revision**
> > >
> > > We sincerely appreciate your positive feedback and the recommendation to accept.
> > >
> > > Per your suggestion, we have clarified the qualitative nature of the recommendations by adding the following to the penultimate paragraph of the *Introduction*:
> > >
> > > >In the absence of broad quantitative consensus, we adopt a robust, evidence-based methodology for making recommendations. By leveraging a deep qualitative analysis of the literature, we derive reliable method comparisons and guidelines that offer essential, practical guidance.
> > >
> > > Thank you again for your time and valuable insights.
> > >
> > > Best regards,
> > >
> > > The Authors

---

### Review · Reviewer_2cbv · 2025-10-13

**Summary Of Contributions:**

The authors provide a very extensive overview of different approaches and uncertainty estimation techniques for semantic segmentation. They formalize how the different approaches model and disentangle uncertainty. They further analyze downstream applications for uncertianty quantification. The paper ends with a comprehensive discussion of the problems and future outlook for various parts of the field.

**Additional Comments:**

Question:
Do the authors believe that the attempts to disentangle uncertainty sources during modeling is hindering progress of using UQ for downstream applications?

**Audience:**

Yes

**Audience Explanation:**

It is a very comprehensive overview of the topic and will be very helpful for everyone who seeks to start working on this topic.

**Broader Impact Concerns:**

No concerns

**Claims And Evidence:**

Yes

**Claims Explanation:**

Comprehensive review covering the main approaches of the field.

**Requested Changes:**

- Some methods could be elaborated a bit more on how they are connected to the current topic: e.g. label smoothing in 4.1.1. is only mentioned after a long background on calibration
- Explain the differences between the entropy decompositions. I really like the fact that this is done, but each time the authors could spend 2-3 sentences more to really explain what is shown and how it differs to the other equations and assumptions
- The applications sections could benefit from a high-level example in the beginning each time: e.g. observer variablility could be made very clear given an example there

Not necessarily requested, but I believe it would help the work:
- Streamlining the paper. Some paragraphs explain very basic background, like GANs, Diffusion models, etc. I could see a more distilled main text while adding this information in the appendix. This could help with a clear focus on the work.

Typos:
- In 5.2 laplace approximation: „.e.g.“
- In 5 parameter modeling last sentence in the overview has the incorrect order of the sections

---

> ### Author Response · Authors · 2025-10-25
> **Response to review**
>
> We thank the reviewer for their supportive feedback and positive evaluation. We are confident that the changes, detailed below, have significantly enhanced the paper's clarity.
>
> **Some methods could be elaborated a bit more on how they are connected to the current topic: e.g. label smoothing in 4.1.1. is only mentioned after a long background on calibration.**
>
> We agree that the connection between calibration methods and the paper's general themes could be improved. To address this, we have comprehensively revised *Section 4.1.1*. First, instead of a long background, the concept of calibration is now introduced concisely within its own paragraph. This provides a clearer foundation to then introduce methods designed to enforce calibration. Crucially, we now explicitly connect these methods to our established framework by integrating them with the notation and equations already present in the paper.
>
> **Explain the differences between the entropy decompositions. I really like the fact that this is done, but each time the authors could spend 2-3 sentences more to really explain what is shown and how it differs to the other equations and assumptions.**
>
> To provide the deeper, comparative analysis the reviewer requested, we have revised our treatment of the entropy decompositions. For each key equation (now *Equations 3, 42, and 43*), we have now added several sentences that explicitly define each term and clarify the underlying assumptions. We are confident this expanded explanation delivers the detailed analysis the reviewer was looking for.
>
> **The applications sections could benefit from a high-level example in the beginning each time: e.g. observer variablility could be made very clear given an example there.**
>
> Following this advice, we have now added a high-level, practical example at the beginning of the sections on *Observer Variability, Active Learning, and Model Introspection* to better orient the reader.
>
> **Streamlining the paper. Some paragraphs explain very basic background,like GANs, Diffusion models, etc. I could see a more distilled main text while adding this information in the appendix. This could help with a clear focus on the work.**
>
> We have condensed the introductory material where possible. We have respectfully chosen to retain the background explanations in the main text for two primary reasons.
>
> 1. A core goal of our work is to be a self-contained and accessible resource, particularly for readers crossing over from different sub-fields. Moving foundational content to an appendix would hinder this accessibility.
> 2. Our introductions are specifically tailored to connect these fundamental methods to the unifying notational and theoretical framework we establish for probabilistic segmentation. This integration is crucial for demonstrating how disparate areas of the field can be viewed through a common lens.
>
> While this background may be familiar to some experts, we are confident that its inclusion is essential for the paper's pedagogical value and its contribution as a cross-domain synthesis.
>
> **Typos.**
>
> Thank you!
>
> **Do the authors believe that the attempts to disentangle uncertainty sources during modeling is hindering progress of using UQ for downstream applications?**
>
> This is an excellent question that gets to the heart of a central debate in the field. Our position, which we have now made explicit in the revised manuscript, is that the pursuit of uncertainty disentanglement should be a means to an end, not the primary goal itself. Our reasoning is based on the following points:
>
> 1. We believe the ultimate goal of uncertainty modeling is the practical utility, where models are made to be more reliable and performant on specific downstream tasks. A model's total uncertainty score could be already sufficient for tasks, making perfect separation unnecessary.
> 2. Demonstrating disentanglement can be a distraction, requiring additional validation problems that divert attention from the core task.
>
> Understanding the reducibility of the uncertainty type helps researchers select better modeling strategies or decide where to acquire new data. However, this diagnostic role should not be mistaken for the ultimate objective. We clarify this position further in *Section 8.1.2,* detailing when the conceptual reducibility of uncertainty is useful. As we argue in *Section 8.4*, the most important qualities of an uncertainty estimate are that it is interpretable and actionable. Consequently, we consider the goal of disentangling uncertainty into its sources to be of secondary importance compared to achieving these practical outcomes.

---

> > ### Comment · Reviewer_2cbv · 2025-11-20
> > **Thanks!**
> >
> > I thank the authors for their rebuttal. Most of my concerns have been addressed. I already gave my positive feedback a while ago.
> >
> > All the best for the submission.

---

### Author Response · Authors · 2025-10-25
**General Comment**

We sincerely thank all reviewers for their thoughtful and constructive feedback. We are delighted that the paper was received so positively and are grateful for the time and expertise dedicated to its review. We're particularly encouraged that the reviewers found our work to be a **comprehensive and extensive overview** that will serve as a **useful resource and reference**. Furthermore, we are pleased that the topic is considered important and **highly relevant to the TMLR audience**, with its clear applications in safety-critical domains. We also deeply appreciate the reviewers' recognition of the manuscript's **overall quality** and **clear presentation**. The suggestions for improvement were equally valuable and have guided a substantial revision, detailed below in our point-by-point response, of the manuscript.

**Fundamental Restructuring and Practical Guidance**

- We addressed a central tension in the manuscript between its taxonomy, which relied on the aleatoric/epistemic distinction, and our own critical discussion of that framework's limitations.
  - We **fundamentally restructured** the manuscript to be organized around empirically grounded categories based on modeling strategy rather than uncertainty type.
  - To improve the manuscript's narrative focus, the **in-depth, critical discussion** of epistemic and aleatoric uncertainty is consolidated into the dedicated subsection (*Section 7.1*), while brief allusions to these concepts are retained in the main text where contextually relevant.
- A key suggestion for improving the manuscript was to better serve practitioners by introducing a more cohesive framework for uncertainty evaluation and by offering clearer, actionable guidance on method selection.
- To holistically address these concerns, we have introduced an **entirely new** capstone *Section 8: "Recommendations and Guidelines".* This section offers actionable support for practitioners, starting with guidance on **method selection, evaluation and reproducibility**, which is visualized in a new decision flowchart (*Figure 15*) and detailed in a summary of comprehensive trade-offs in the expanded *Table 7*. Furthermore, it provides a crucial critical perspective by **addressing the limitations** of the selection process and **establishing clear criteria** for what constitutes trustworthy uncertainty.

**Enhanced Critical Analysis and Quantitative Comparison**

- The manuscript would be strengthened by a more structured critical analysis of method effectiveness and a deeper discussion of method failures.
  - To provide more balanced analysis, we **expanded** *Section 7.5: "Method suboptimality"*, and added Section 8.1 "Method selection", which is dedicated to discussing method failures and trade-offs across applications.
  - We have **added quantitative data** where possible, most notably by **significantly expanding** *Table 7* to compare all surveyed methods across more concrete metrics

**Improved Narrative Flow and Clarity**

- The manuscript's impact could be enhanced by improving the narrative cohesion between sections to better guide the reader through the paper's central argument and key contributions.
  - To improve the narrative flow, we **added guiding text at critical junctures**, including **a new roadmap** *Section 3.2: "Review Organization and Structure"*, and revised introductory paragraphs for all major sections.
  - We **integrated** "*Key Takeaways*" boxes and **summarizing bullet points** throughout the manuscript to help orient the reader.
- The paper's contribution could be sharpened by more explicitly defining its target audience and unique positioning within the literature.
  - To clarify the paper's contribution and audience, we **modified the Introduction** framing the work as a "*cross-domain synthesis*" for researchers, rather than a standard introduction.

**Elaboration on Specific Concepts**

- We acted on the reviewers' feedback to provide more detailed explanations for several key concepts.
  - We **restructured** *Section 4.1.1* to more **clearly connect** label smoothing to calibration and the paper's overall framework.
  - We added several sentences of **detailed explanation** for each equation related to the entropy decompositions to clarify assumptions and differences.
  - We added high-level, **practical examples** at the beginning of Sections 6.1-6.3*.
  - We **revised** the caption and content of *Figure 11* to better illustrate the comparison between different kernel sampling techniques.
  - Finally, we have **corrected all** identified typos, grammatical errors, and citation formatting issues throughout the manuscript.

We wish to thank the assigned Action Editor for managing the review process, and the reviewers for their insightful and constructive feedback. We have diligently addressed all comments and believe the revised manuscript is substantially improved. We look forward to your assessment.



Sincerely,

The authors,

---

### Decision · Action_Editor_5isK · 2025-11-30

**Recommendation:** Accept with minor revision

**Additional Comments:**

There remain a few minor issues in the presentation. For example:

In Table 5, the “Various CT” row contains incomplete references with question marks (e.g., “Hu et al. (2023;?)”, “LaBonte et al. (2019;?)”).

On p.36, “Mutli Label” should be corrected to “Multi-Label.”

The authors are encouraged to carefully proofread the manuscript and resolve these and other typographical or formatting inconsistencies before final submission.

Also, the reviewers suggest that the limitations concerning the lack of standardized benchmarks and comprehensive empirical validation, which are currently detailed in Section 8.3, should be clearly stated upfront in the abstract and/or introduction.

**Audience:**

Yes

**Audience Explanation:**

The manuscript offers a comprehensive survey of uncertainty quantification in deep semantic segmentation, a topic of clear relevance and interest to the computer vision and machine learning community.

**Claims And Evidence:**

Yes

**Claims Explanation:**

The paper provides an extensive and thorough overview of uncertainty quantification methods in semantic segmentation, synthesizing a diverse and rapidly expanding body of work. The reviewers highlight several key strengths: (1) the paper effectively organizes a fragmented literature through a clear taxonomic structure and maintains a solid overall presentation quality; and (2) it identifies important practical challenges and open research directions, making it a useful reference for research communities.

Although the initial reviews raised several concerns—such as insufficient empirical validation, ambiguities in the proposed uncertainty framework, unclear structure and motivation, and limited practical guidance—the authors’ rebuttal and revisions have substantially improved the manuscript. Following the discussion, all reviewers now recommend acceptance, noting that the authors have adequately addressed the major issues and strengthened the paper’s central claims. Reviewer HJet requests one remaining minor clarification concerning the articulation of current limitations.

The AE concurs with the reviewers’ consensus and recommends acceptance. The authors are encouraged to incorporate the remaining clarification and address any residual concerns in the final camera-ready version.

---

> ### Author Response · Authors · 2025-12-08
> **Thank you**
>
> We have carefully re-read the manuscript and made the following edits:
> - removed incomplete references in Table 5
> - hyphenated "Multi-Label"
> - added sentences to the penultimate paragraph in the introduction:
> > In the absence of broad quantitative consensus, we adopt a robust, evidence-based methodology for making recommendations. By leveraging a deep qualitative analysis of the literature, we derive reliable method comparisons and guidelines that offer essential and practical guidance. However, this reliance on qualitative synthesis rather than quantitative evidence is an important limitation and our recommendations should be interpreted accordingly.
>
> Thank you again for your positive evaluation. Our camera-ready submission has been uploaded.